# PURA syndrome-causing mutations impair PUR-domain integrity and affect P-body association

**Marcel Proske[1,2†], Robert Janowski[1†], Sabrina Bacher[1], Hyun-Seo Kang[1,3], Thomas Monecke[2], Tony Koehler[2], Saskia Hutten[4], Jana Tretter[1], Anna Crois[2], Lena Molitor[1], Alejandro Varela-Rial[5], Roberto Fino[5], Elisa Donati[5], Gianni De Fabritiis[5], Dorothee Dormann[4,6], Michael Sattler[1,3], Dierk Niessing[1,2]\***

[1]Institute of Structural Biology, Molecular Targets and Therapeutics Center, Helmholtz Munich, Neuherberg, Germany; [2]Institute of Pharmaceutical Biotechnology, Ulm University, Ulm, Germany; [3]Chemistry Department, Biomolecular NMR and Center for Integrated Protein Science Munich, Technical University of Munich, Mainz, Germany; [4]Biocenter, Institute of Molecular Physiology, Johannes Gutenberg-Universität (JGU), Mainz, Germany; [5]Acellera Labs SL, Barcelona, Spain; [6]Institute of Molecular Biology (IMB), Mainz, Germany

**\*For correspondence:**
dierk.niessing@uni-ulm.de

[†]These authors contributed equally to this work

**Competing interest:** The authors declare that no competing interests exist.

**Abstract** Mutations in the human *PURA* gene cause the neurodevelopmental PURA syndrome. In contrast to several other monogenetic disorders, almost all reported mutations in this nucleic acid-binding protein result in the full disease penetrance. In this study, we observed that patient mutations across PURA impair its previously reported co-localization with processing bodies. These mutations either destroyed the folding integrity, RNA binding, or dimerization of PURA. We also solved the crystal structures of the N- and C-terminal PUR domains of human PURA and combined them with molecular dynamics simulations and nuclear magnetic resonance measurements. The observed unusually high dynamics and structural promiscuity of PURA indicated that this protein is particularly susceptible to mutations impairing its structural integrity. It offers an explanation why even conservative mutations across PURA result in the full penetrance of symptoms in patients with PURA syndrome.

## eLife assessment

This **important** study addresses the mechanisms by which mutations in the PURA protein, a regulator of gene transcription and mRNA transport and translation, cause the neurodevelopmental PURA syndrome. Based on **convincing** evidence from structural biology, molecular dynamics simulation, biochemical, and cell biological analyses, the authors show that the PURA structure is very dynamic, rendering it generally sensitive to structure-altering mutations that affect its folding, DNA-unwinding activity, RNA binding, dimerization, and partitioning into processing bodies. These findings are of substantial importance to cell biology, neurogenetics, and neurology alike, because they provide first insights into how very diverse PURA mutations can cause similar and penetrant molecular, cellular, and clinical defects.

## Introduction

The family of purine-rich element-binding (PUR) protein is highly conserved from plants to humans (*Molitor et al., 2021*). Among the three vertebrate paralogs PURA, PURB, and PURG, the best-studied

**eLife digest** PURA syndrome is a neurodevelopmental disorder that affects about 650 patients worldwide, resulting in a range of symptoms including neurodevelopmental delays, intellectual disability, muscle weakness, seizures, and eating difficulties.

The condition is caused by a mutated gene that codes for a protein called PURA. PURA binds RNA – the molecule that carries genetic information so it can be translated into proteins – and has roles in regulating the production of new proteins. Contrary to other conditions that result from mutations in a single gene, PURA syndrome patients show 'high penetrance', meaning almost every reported mutation in the gene leads to symptoms.

Proske, Janowski et al. wanted to understand the molecular basis for this high penetrance. To find out more, the researchers first examined how patient mutations affected the location of the PURA in the cell, using human cells grown in the laboratory. Normally, PURA travels to P-bodies, which are groupings of RNA and proteins involved in regulating which genes get translated into proteins. The researchers found that in cells carrying PURA syndrome mutations, PURA failed to move adequately to P-bodies.

To find out how this 'mislocalization' might happen, Proske, Janowski et al. tested how different mutations affected the three-dimensional folding of PURA. These analyses showed that the mutations impair the protein's folding and thereby disrupt PURA's ability to bind RNA, which may explain why mutant PURA cannot localize correctly.

Proske, Janowski et al. describe the molecular abnormalities of PURA underlying this disorder and show how molecular analysis of patient mutations can reveal the mechanisms of a disease at the cell level. The results show that the impact of mutations on the structural integrity of the protein, which affects its ability to bind RNA, are likely key to the symptoms of the syndrome. Additionally, their approach used establishes a way to predict and test mutations that will cause PURA syndrome. This may help to develop diagnostic tools for this condition.

family member is PURA (*Bergemann and Johnson, 1992*; *Haas et al., 1993*; *Haas et al., 1995*). It is ubiquitously expressed and has been implicated in several cellular processes, including transcriptional and translational gene regulation as well as mRNA transport in neurons (*Chepenik et al., 1998*; *Gallia et al., 2001*; *Haas et al., 1993*; *Haas et al., 1995*; *Johnson et al., 2006*; *Kobayashi et al., 2000*; *Mitsumori et al., 2017*; *Tretiakova et al., 1999*). Results from two independent knock-out mouse models demonstrated that PURA is important for postnatal brain development (*Hokkanen et al., 2012*; *Khalili et al., 2003*).

Accordingly, PURA has been implicated as a modulator of neurodegenerative disorders, such as fragile X-associated tremor/ataxia syndrome (FXTAS), and the amyotrophic lateral sclerosis (ALS) frontotemporal dementia spectrum disorder (*Swinnen et al., 2020*). PURA was reported to be present in the pathological RNA foci of both disorders and to protect against RNA toxicity upon ectopic over-expression (*Mori et al., 2013*; *Rossi et al., 2015*; *Shen et al., 2018*; *Swinnen et al., 2018*; *Xu et al., 2013*). In addition, PURA was shown to co-localize with an ALS-causing variant of the FUS protein in stress granules of ALS patients. Importantly, overexpression of PURA reduced the toxicity of mutant FUS by preventing its mis-localization (*Daigle et al., 2016*).

In 2014, two studies reported that a monogenetic neurodevelopmental disorder is caused by sporadic mutations in the *PURA* gene (*Hunt et al., 2014*; *Lalani et al., 2014*). Hallmarks of this so-called PURA syndrome are developmental delay, moderate to severe intellectual disability, hypotonia, epileptic seizures, and feeding difficulties, among others (*Reijnders et al., 2018*; *Johannesen et al., 2021*). Mutations causing PURA syndrome are often frame-shift events but also point mutations distributed over the entire sequence.

Based on the crystal structures of PURA from *Drosophila melanogaster* (*dm*PURA), three conserved sequence regions termed PUR repeats I, II, and III were identified to fold into globular domains (*Graebsch et al., 2010*; *Graebsch et al., 2009*; *Weber et al., 2016*). Whereas PUR repeat I and II assemble into an N-terminal PUR domain via intramolecular interactions, two C-terminal PUR repeats III from different molecules interact with each other and thus mediate dimerization of PURA. All these PUR domains belong to the class of the PC4-like protein family, which mainly bind single-stranded

nucleic acids and can unwind double-stranded DNA and RNA (*Janowski and Niessing, 2020*). Although both PUR domains possess the typical PC4-like β-β-β-β-α-(linker)-β-β-β-β-α topology, the N-terminal domain has been suggested to be the main RNA/DNA interaction entity (*Weber et al., 2016*). In a previous study, homology models based on the *dm*PURA crystal structures (*Graebsch et al., 2009*; *Weber et al., 2016*) were used to predict in silico the effects of PURA syndrome-causing mutations on the structural integrity of the human PURA protein (*Reijnders et al., 2018*). While these mutations could be classified into groups that likely either do or do not impair the structural integrity of PURA, these predictions remained speculative (*Reijnders et al., 2018*). Surprisingly, in contrast to phenotypically related genetic diseases such as the Rett syndrome (*Lombardi et al., 2015*), no hot-spot regions could be identified in the protein sequence that trigger the PURA syndrome. With the exception of the unstructured N-terminal region and the very C-terminus, almost all mutations across the protein sequence appear to result in the full disease spectrum (*Johannesen et al., 2021*; *Reijnders et al., 2018*). To date, we fail to understand why there is such an underrepresentation of mild phenotypes in patients with PURA syndrome.

In this study, we experimentally assessed the structure and function of human PURA as well as the impact of representative PURA syndrome-causing mutations on the protein's integrity. When studying the impact of patient mutations on PURA's subcellular localization, we observed impaired processing body (P-body) association but close-to normal localization to stress granules, indicating a potential importance of P-bodies for PURA syndrome pathology. Two particularly interesting disease-causing variants, K97E and R140P, were further analyzed by structural means. The results indicated that the N-terminal PUR domain is unusually flexible in its fold and prone to misfolding upon mutation. In summary, we provide a structure-based explanation for the underrepresentation of genetic variations with mild symptoms in patients and a rational approach for predicting and functionally testing genetic variations of uncertain significance.

## Results

### Effect of the mutations in *hs*PURA on stress-granule association

Several patient-related genetic variations are frame-shift events and thus predictably destroy the integrity of PURA. In addition, a number of point and indel mutations have been reported for which the pathology-causing effects are less easy to predict (*Johannesen et al., 2021*; *Reijnders et al., 2018*). To understand their functional impact, we decided to first study the effect of three known patient variations, K97E, I206F, and F233del (*Hunt et al., 2014*; *Lalani et al., 2014*; *Reijnders et al., 2018*; *Figure 1A*), on the subcellular localization of *hs*PURA.

Previous studies in human cell cultures indicated that *hs*PURA localizes to stress granules upon cellular stress and may be essential for their formation (*Daigle et al., 2016*; *Markmiller et al., 2018*). To analyze whether the afore-mentioned patient-related genetic variations affect the localization of *hs*PURA to stress granules, the mutant and wild-type versions of full-length FLAG-tagged *hs*PURA were overexpressed in HeLa cells using a doxycycline-inducible expression cassette and cells were stressed for 1 hr with 500 μM sodium arsenite (*Figure 1—figure supplement 1* and *Figure 1B*). To find out whether stress-granule localization of *hs*PURA depends on its nucleic acid-binding properties, we also utilized an RNA-binding-deficient version of *hs*PURA bearing 11 structure-guided point mutations (m11) as a control (for rationale and design of this mutant, see *Figure 1—figure supplement 2*). Using an anti-FLAG tag antibody, we observed wild-type *hs*PURA to be distributed within the cytoplasm and accumulated in stress-granule structures, as seen by co-staining with the stress-granule marker G3BP1 (*Figure 1B*). In contrast, the nucleic acid-binding-deficient mutant *hs*PURA m11 failed to co-localize to G3BP1-positive granules, indicating that RNA binding by *hs*PURA is necessary for its co-localization with stress granules. Furthermore, all three versions of *hs*PURA bearing patient-derived genetic variations localized to stress granules upon cellular stress (*Figure 1B*). In all three patient-related variants, no significant reduction of PURAs stress-granule association was seen when compared to the wild-type control (*Figure 1C*).

To assess the previously reported influence of *hs*PURA on stress-granule formation at endogenous PURA levels (*Daigle et al., 2016*), we performed a small interfering (si) RNA mediated knockdown of *hs*PURA in HeLa cells and stained with the stress-granule marker G3BP1 and PURA. Efficient knockdown of endogenous PURA levels was validated with western blot experiments, showing a significant

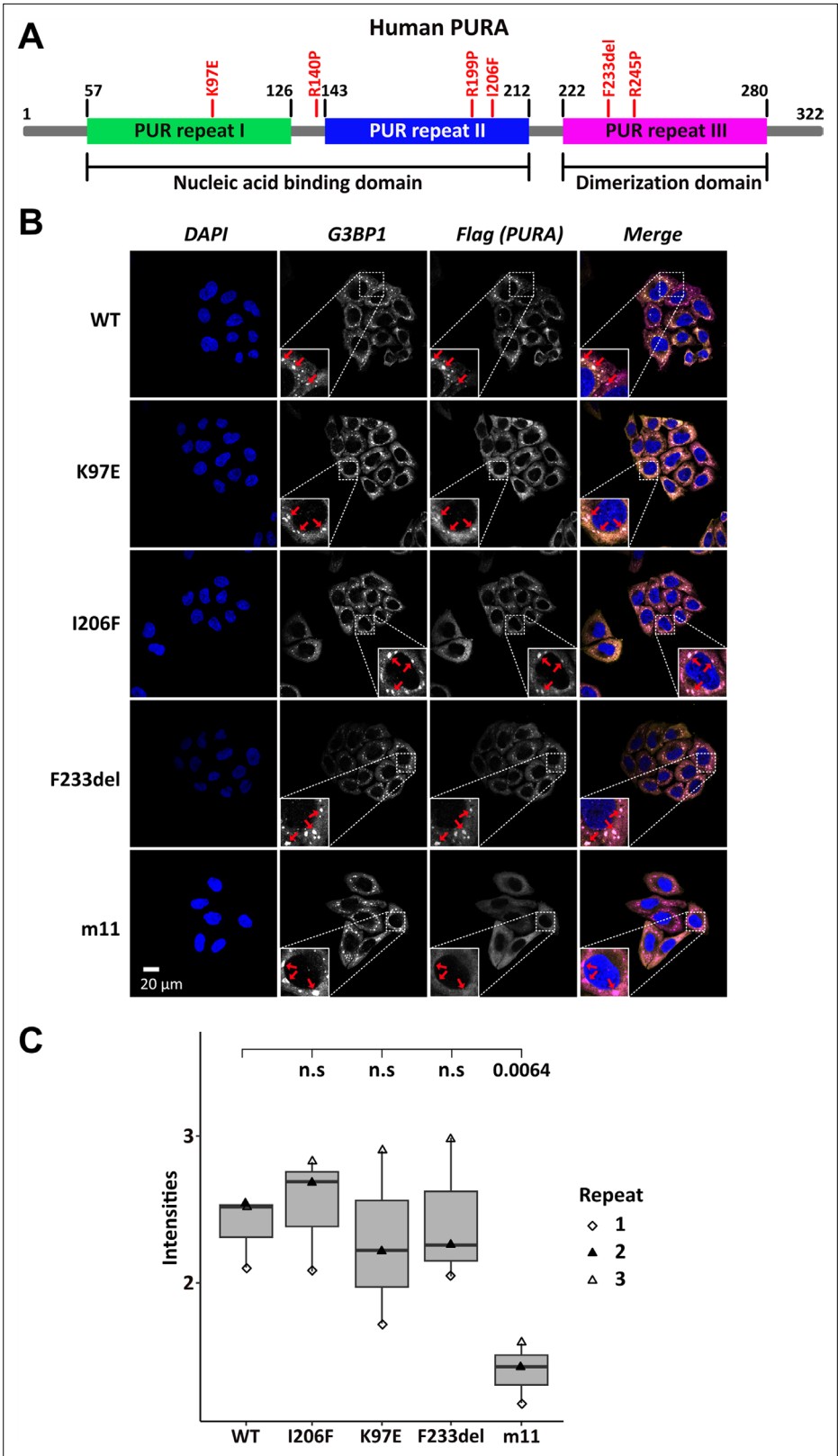

**Figure 1.** Stress-granule localization of *hs*PURA depends on its ability to bind RNA. (**A**) Schematic overview of the *hs*PURA protein. Patient-derived mutations experimentally assessed in this study are marked in red. (**B**) In HeLa cells stress granules were identified by G3BP1 immunostaining (magenta) and overexpressed *hs*PURA (yellow) by its N-terminal Flag-tag. Nuclei were stained with DAPI (4',6-diamidino-2-phenylindole; blue). Representative

*Figure 1 continued on next page*

*Figure 1 continued*

stress granules are marked with red arrows in the zoomed-in area. Except for *hs*PURA m11, all overexpressed versions of *hs*PURA (wild-type and mutant) showed strong stress-granule localization. (**C**) Box–Whiskers plot of quantified intensities of *hs*PURA staining in stress granules above cytoplasmic background levels upon arsenite-induced stress treatment (see **A**). Three biological replicates (indicated by different icons) with 66 stress granules each were quantified for each cell line. Box–Whiskers plot of average signal intensity per repeat in each cell line. No significant difference was detected between patient-related genetic variations and wild-type (WT). Only for the m11 mutant protein the association to stress granules was significantly reduced (p = 0.0064), indicating the requirement of RNA binding for stress-granule association. p-value was calculated using unpaired two-sided Student's *t*-test of mutated protein compared to WT control.

The online version of this article includes the following source data and figure supplement(s) for figure 1:

**Figure supplement 1.** Quantitative western blot of cell lysate of cell lines used for the immunofluorescence (IF) staining assays.

**Figure supplement 1—source data 1.** Quantitative western blot of cell lysate of cell lines used for the immunofluorescence staining assays – uncropped, raw image.

**Figure supplement 1—source data 2.** Quantitative western blot of cell lysate of cell lines used for the immunofluorescence staining assays – uncropped, labeled image.

**Figure supplement 2.** RNA-binding-deficient *hs*PURA I–II m11 variant.

**Figure supplement 3.** Validation of knockdown efficiency.

**Figure supplement 3—source data 1.** Validation of knockdown efficiency.

**Figure supplement 3—source data 2.** Validation of knockdown efficiency.

Western blot of control (scrambled sirRNA) and PURA knockdown HeLa cells anti-PURA and H3 as housekeeping gene – uncropped, labeled images.

**Figure supplement 4.** Stress-granule formation in HeLa wild-type versus PURA knockdown cells.

reduction of PURA protein upon knockdown (*Figure 1—figure supplement 3*). As expected, endogenous *hs*PURA co-localized with G3BP1 upon stress treatment with 500 µM arsenite PURA (*Figure 1—figure supplement 4A, B*) and thus behaved similar to overexpressed *hs*PURA. A PURA knockdown did not alter the total number of microscopically visible stress granules but resulted in a greater ratio of smaller stress granules (*Figure 1—figure supplement 4C*). Although this finding suggests that PURA is not essential for stress-granule formation, it should be noted that the knockdown did not completely eliminate PURA levels and hence a stronger effect on stress granules could occur in a complete knock-out of PURA. However, PURA syndrome patients bear a heterozygous mutation, suggesting that an incomplete knockdown likely resembles the pathogenic situation more closely than a full knock-out. Together these findings suggest that (1) reduced stress-granule association of functionally impaired *hs*PURA variants may not be a core feature of the PURA syndrome and (2) impaired stress-granule formation is not very likely to be directly responsible for the etiology of this disorder.

## Localization of *hs*PURA to P-bodies is impaired by PURA syndrome-causing mutations

To assess the localization of *hs*PURA to cytoplasmic granules in unstressed conditions, we first recapitulated its recently reported localization to P-bodies (*Molitor et al., 2023*). Co-staining with the marker DCP1A indeed confirmed a co-localization of overexpressed *hs*PURA in P-bodies of HeLa cells (*Figure 2A, B*). As for stress granules, the PURA m11 mutant failed to accumulate in P-bodies, indicating that RNA binding of PURA contributes to its P-body localization. In addition, the patient-related genetic variants K97E and F233del of *hs*PURA showed significantly reduced co-localization to P-bodies (*Figure 2A, B*). In summary, these experiments indicate that P-body localization of *hs*PURA requires its RNA-binding activity, and that the majority of tested patient variations result in impaired P-body localization. Furthermore, since F233del has been predicted to have impaired dimerization properties (*Reijnders et al., 2018*), these findings suggest dimerization to be important for PURA's P-body association.

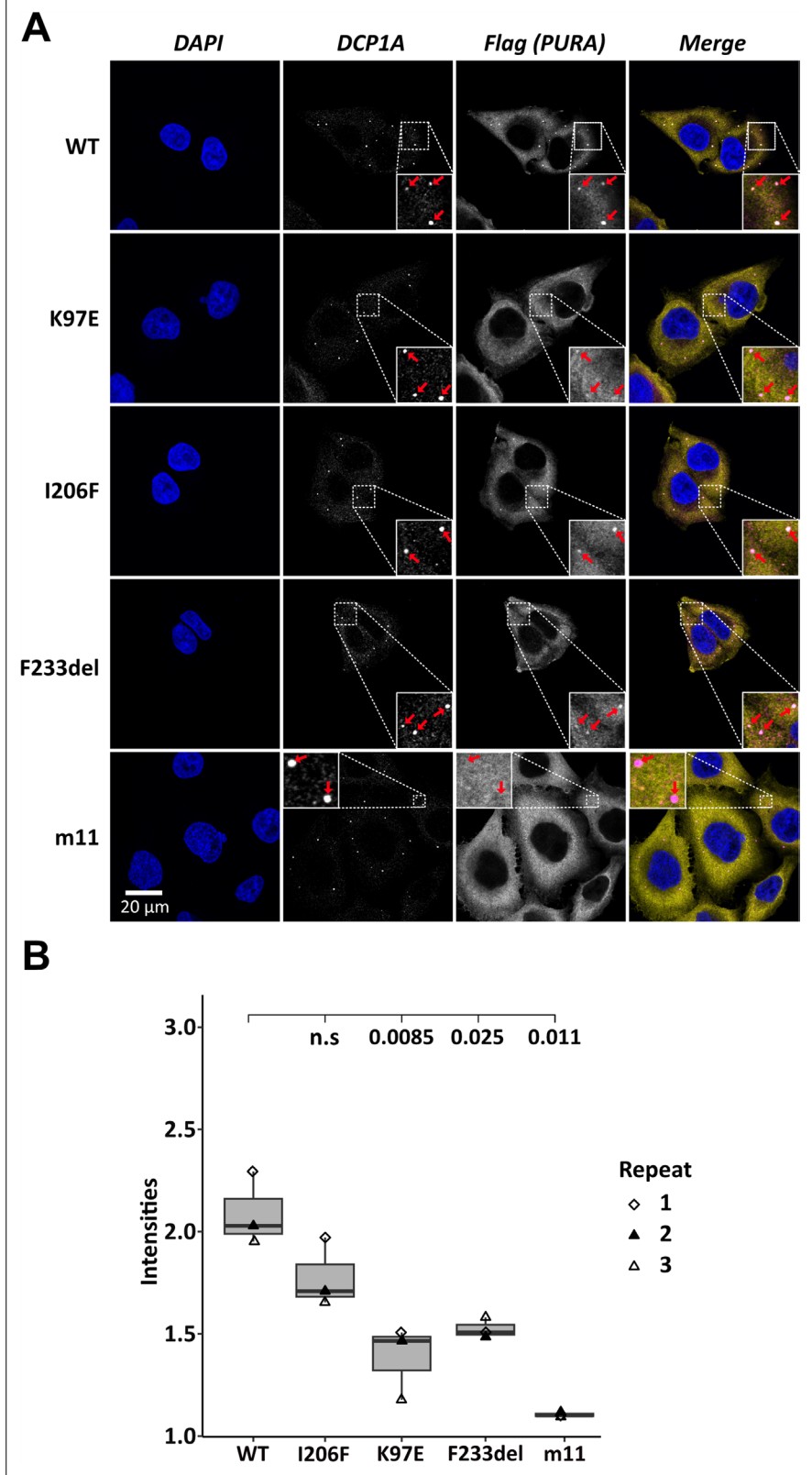

**Figure 2.** Processing-body association of *hs*PURA in HeLa cells is impaired by patient-derived mutations. (**A**) P-bodies were identified by immunostaining against DCP1A and against the N-terminal FLAG-tag of overexpressed *hs*PURA FL. Whereas their individual staining is shown in white, the overlay of DCP1A and FLAG is shown in magenta and yellow, respectively. Nuclei were stained with DAPI (blue). All cell lines show P-body formation,

*Figure 2 continued on next page*

*Figure 2 continued*

marked with red arrows in the zoomed area. (**B**) Box–Whiskers plots of quantification of intensities of *hs*PURA staining in P-bodies above cytoplasmic background levels. For each cell line, three biological replicates (indicated by different icons) with each 100 P-bodies were quantified, and Box–Whiskers plot generated for each cell line. Significantly lower P-body localization was observed for the K97E (p = 0.0085), F2333del (p = 0.025), and m11 (p = 0.011) variants of *hs*PURA. p-values were calculated using unpaired two-sided Student's *t*-test of mutant compared to WT control.

The online version of this article includes the following figure supplement(s) for figure 2:

**Figure supplement 1.** *hs*PURA does not undergo phase separation.

## *hs*PURA does not readily undergo RNA-driven phase separation in vitro

*hs*PURA accumulates in an RNA-dependent fashion to stress granules as well as to P-bodies. Both types of granules are liquid-phase-separated entities, indicating that the recruitment to such granules might occur via phase separation of RNA-bound *hs*PURA. Since *hs*PURA was recently shown to be required for P-body formation in HeLa cells and fibroblasts (*Molitor et al., 2023*), PURA-dependent liquid-phase separation could potentially also directly contribute to the formation of these granules. Furthermore, a loss of such a function could potential contribute to the etiology of PURA syndrome. In order to test this hypothesis, we used a microscope-based approach to visualize the ability of RNA-bound *hs*PURA to phase separate in vitro. As a positive control for RNA-mediated liquid-phase separation, the protein FUS RGG3 was used (*Hofweber et al., 2018*). In contrast to this control, no indication of phase separation was observed for *hs*PURA either in the presence or absence of total HeLa cell RNA (*Figure 2—figure supplement 1*). These results indicate that *hs*PURA does not readily phase separate with total cellular RNA under the tested experimental conditions. Of note, this observation does not exclude the possibility that *hs*PURA can undergo phase separation under different experimental conditions, for instance in presence of additional co-factors. However, when putting this observation in perspective with previous reports, it seems unlikely that P-body formation directly depends on phase separation by *hs*PURA, but rather on its recently reported function as gene regulator of the essential P-body core factors LSM14a and DDX6 (*Molitor et al., 2023*).

## Recombinant expression tests of PURA protein with disease-causing mutations

To understand the impact of mutations on the molecular and structural integrity of *hs*PURA, we recombinantly expressed the above-described *hs*PURA proteins K97E, I206F, and F233del in *E. coli*. In addition, *hs*PURA with either of the mutations R140P, R199P, and R245P were recombinantly expressed (*Figure 1A*). All three mutations exchange an arginine against a proline, albeit in different positions of the protein and very different structural contexts. R140P was initially reported to be disease-causing (*Lee et al., 2018*) but later re-interpreted as a genetic variation for which it is unclear if it can trigger PURA syndrome, that is of uncertain significance (https://www.ncbi.nlm.nih.gov/clinvar/variation/192343/). In contrast, the genetic variations R199P and R245P have been described as disease causing, with matching symptoms for patients with PURA syndrome (*Reijnders et al., 2018*). Hence, R140P was chosen in this study to explore the diagnostic potential of our approach, whereas R199P and R245P served as a reference point for pathology-inducing genetic variations.

Even after extensive optimization, recombinantly expressed *hs*PURA fragments with the mutations R199P and I206F (full-length variant and N-terminal PUR domain consisting of repeats I–II; PURA I–II) as well as F233del and R245P (full-length variant and C-terminal PUR domain consisting of two repeats III; PURA III) could not be purified due to protein insolubility (*Supplementary file 1a*). In contrast, N-terminal PUR domains (PURA I–II) with either of the two remaining mutations K97E and R140P were soluble and could be purified to near homogeneity. The observed solubility of recombinant *hs*PURA I–II K97E is also in agreement with the previous in silico prediction (*Reijnders et al., 2018*) as K97 has a surface-exposed side chain that is unlikely to cause folding issues upon mutation into another flexible, polar side chain. For the R140P variant with unclear pathological significance, no conclusive prediction on its impact on protein folding could be made in the past.

## The C-terminal dimerization PUR domain adopts a classical PC4-like fold

Several PURA syndrome-causing genetic variations have been reported to be located in the C-terminal, dimerization-mediating PUR domain (*Johannesen et al., 2021*; *Reijnders et al., 2018*). Due to the lack of an experimental structure of human PURA, previous predictions of the impact of patient variants on the structural integrity of *hs*PURA had to rely on a human homology model derived from *dm*PURA (*Reijnders et al., 2018*). Since *Drosophila* and human PURA have a rather moderate sequence identity of 46%, we decided to establish a better experimental basis for predicting the structural effects of patient-derived genetic variations and determined the crystal structure of the fragment P215-K280 (i.e. *hs*PURA III) at 1.7 Å resolution (*Figure 3A* and *Supplementary file 1b*). This fragment corresponds to the domain boundaries suggested by the previously solved invertebrate *dm*PURA III structure (*Weber et al., 2016*). The structure of the C-terminal PUR-domain possesses the typical four antiparallel β-strands and one α-helix. Like in the previously published *dm*PURA III structure (*Weber et al., 2016*), the C-terminal PUR domain of *hs*PURA is built from two PUR repeats III of independent protein chains that interact with each other to form an intermolecular homodimer. The *hs*PURA III dimer shows high structural similarity to *dm*PURA III with root mean square deviation (r.m.s.d.) of 1.26 Å for 125 superimposed Cα atoms and 50% of sequence identity (*Figure 3—figure supplement 1A*).

## Dimerization domain of *hs*PURA shows nucleic acid binding and unwinding activities

PUR repeats III adopt a classical PC4-like fold, suggesting that it might also bind to nucleic acids. Analysis of the crystal structure of the *hs*PURA III homodimer revealed electrostatic surface potentials favorable for interactions with nucleic acids (*Figure 3—figure supplement 1B*). When performing gel electrophoresis mobility shift assays (EMSAs) we indeed observed RNA binding (*Figure 3B, C*).

It had been previously reported that PURA can separate double-stranded (ds) nucleic acids in an ATP-independent fashion (*Darbinian et al., 2001*; *Weber et al., 2016*). This so-called unwindase activity has also been reported for several other PC4-like domains from different organisms (*Janowski and Niessing, 2020*). For *dm*PURA it had already been suggested that dsDNA strands are melted by intercalation of the β-ridge between both strands, followed by binding of both single strands to the cavities formed by the curved β-sheets on both sides of the β-ridge (*Figure 3—figure supplement 2*; *Weber et al., 2016*). Since we had already observed RNA binding by PUR repeats III (*Figure 3B, C*), we examined its potential unwindase activity by utilizing a previously reported in vitro unwinding assay (*Darbinian et al., 2001*; *Weber et al., 2016*) with dsDNA 5'-end labeled with a FAM fluorophore and 3'-end labeled with Dabcyl quencher (*Figure 3—figure supplement 3*). In this assay, double-stranded DNA results in low fluorescence signal due to the proximity of the quencher to the fluorophore. Upon strand separation their distance and fluorescence signal increase. We indeed observed strand separation in presence of the *hs*PURA III fragment (*Figure 3D*). Of note, RNA binding as well as dsDNA strand separation by full-length *hs*PURA was much more efficient than by the PUR III domain (*Figure 3D, E*), indicating potential cooperativity of the N- and C-terminal PUR domains in unwinding double-stranded nucleic acids.

## PURA syndrome-causing genetic variations in the C-terminal PUR domain impair dimerization

Next, we used the *hs*PURA III structure (*Figure 3A* and *Supplementary file 1b*) to perform an in silico analysis of the impact of patient-derived variants F233del and R245P on the structural integrity of the C-terminal PUR domain. While failure to obtain soluble recombinant PURA with F233del prevented a direct biophysical assessment of its dimerization state in vitro (*Supplementary file 1a*), this observation already indicated a misfolding of the C-terminal PUR domain upon mutation. Consistent with this observation is also the interpretation from homology modeling based on the previously published *dm*PURA III structure (*Reijnders et al., 2018*). There, the mutation was predicted to impair the structural integrity of the C-terminal *hs*PURA dimerization domain. F233 is part of the β-sheet and interacts with the α-helix of the other PUR repeat III within the PUR domain, contributing to dimerization (*Figure 3—figure supplement 4A*). Since in the PURA F233del variant, one residue is deleted, the orientation of all subsequent amino acids in the β-strand

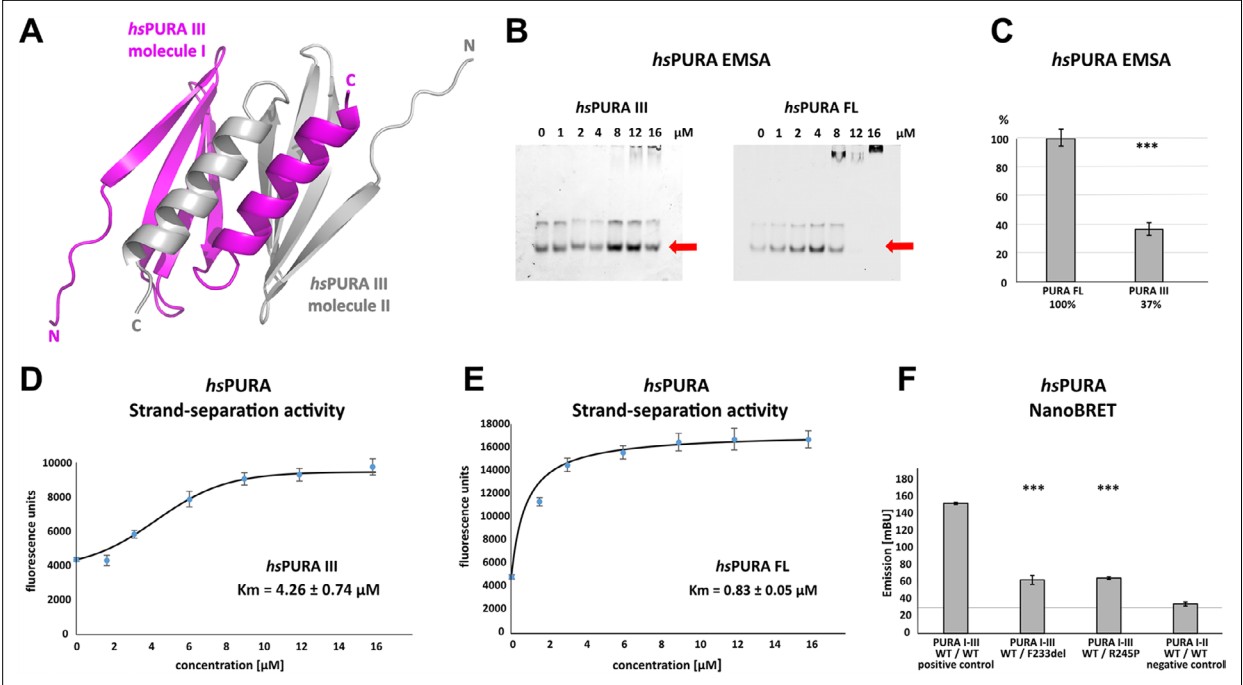

**Figure 3.** Crystal structure of C-terminal PUR domain from *hs*PURA and effects of mutations on its function. (**A**) Crystal structure of *hs*PURA III at 1.7 Å resolution (PDB ID: 8CHW). Two *hs*PURA III molecules (magenta and gray, respectively) form a homodimer. Like other PUR domains (e.g. *hs*PURA I–II), *hs*PURA repeat III consists of four β-sheets and one α-helix. (**B**) In electrophoretic mobility shift assays (EMSAs) the interaction of *hs*PURA FL and *hs*PURA III with a 24-mer (CGG)$_8$ RNA fluorescently labeled with Cy5 fluorophore at 5'-end was observed. The amount of the RNA was kept constant at 8 nM while the protein concentration increased from 0 to 16 μM. The unbound RNA used for quantification (see E) is indicated with red arrows. (**C**) Apparent affinities derived from EMSAs (**C**) indicate that the C-terminal PUR domain is also able to interact with the nucleic acids. Pairwise *t*-tests of the *hs*PURA fragments showed significantly lower RNA binding compared to the full-length protein (*hs*PURA III: p = 9.6E−4). Three replicates were measured for each experiment and protein variant, and the standard deviations have been calculated and shown as bars. (**D, E**) Strand-separation activity of *hs*PURA. The graphs show the averaged values of three independent experiments as dots with standard deviations as error bars. (**D**) The *hs*PURA III fragment shows sigmoidal increase of the ssDNA concentration measured as a fluorescent signal. For the quantification of the sigmoidal curves, we utilized Boltzmann function implemented in the Origin software. Calculated $x_0$, which corresponds to the Km in this assay yields 4.26 ± 0.74 μM. (**E**) Strand-separation activity of full-length *hs*PURA was calculated with Michaelis–Menten kinetic, yielding an average Km value of 0.83 ± 0.05 μM, respectively. For both *hs*PURA samples at least three independent measurements have been performed. (**F**) NanoBRET experiments in HEK293 cells with different *hs*PURA fragment-expressing constructs. Milli-BRET units (mBU) were measured for dimerization of *hs*PURA I–III with *hs*PURA I–III, *hs*PURA I–III F233del, and *hs*PURA I–III R245P. Pairwise *t*-tests of the mutant *hs*PURA I–III versions F233del and R245P yielded significantly lower signals compared to the wild-type protein (*hs*PURA I–III F233del: p = 3.3E−4; *hs*PURA I–III R245P: p = 1.5E−7), indicating impaired interactions between the proteins. Black horizontal line shows reference of mBU obtained for *hs*PURA I–II as negative control. Of note, since the BRET signal is the ratio of donor and acceptor signal, it does not require normalization for expression levels. Asterisks in (**C**) and (**F**) indicate significance level: *** for p ≤ 0.001.

The online version of this article includes the following source data and figure supplement(s) for figure 3:

**Source data 1.** Electrophoretic mobility shift assays (EMSAs) for the interaction of *hs*PURA variants with a 24-mer (CGG)$_8$ RNA fluorescently labeled with Cy5 fluorophore at 5'-end.

**Source data 2.** Electrophoretic mobility shift assays (EMSAs) for the interaction of *hs*PURA variants with a 24-mer (CGG)$_8$ RNA fluorescently labeled with Cy5 fluorophore at 5'-end.

**Source data 3.** Electrophoretic mobility shift assays (EMSAs) for the interaction of *hs*PURA variants with a 24-mer (CGG)$_8$ RNA fluorescently labeled with Cy5 fluorophore at 5'-end.

**Source data 4.** Electrophoretic mobility shift assays (EMSAs) for the interaction of *hs*PURA variants with a 24-mer (CGG)$_8$ RNA fluorescently labeled with Cy5 fluorophore at 5'-end.

**Figure supplement 1.** Analysis and comparison of the crystal structure of *hs*PURA repeat I-II and III.

**Figure supplement 2.** Structural features likely contributing to the PURA-dependent unwinding of dsRNA.

**Figure supplement 3.** The strand-separation activity assay DNA probe.

**Figure supplement 4.** The crystal structure of the *hs*PURA III homodimer.

adopt the opposite orientation, likely resulting in a fold disruption of the β-sheet and possibly in impaired dimerization. In the case of the R245P variant (*Figure 3—figure supplement 4B*), the exchange into proline also likely impairs β-strand folding by introducing a kink, which could also impact dimerization.

To provide direct proof for impaired dimerization upon the introduction of patient-derived F233del and R245P variants, we utilized the FRET-based NanoBRET assay in HEK293 cells (*Figure 3F*). With this assay, we assessed the ability of *hs*PURA bearing the F233del and R245P patient variants to dimerize with wild-type *hs*PURA by measuring corresponding milli BRET Units (mBU). Of note, this combination of wild-type and mutated PURA protein is likely to recapitulate the heterozygous situation in patients. As a control for background signal, non-dimerizing *hs*PURA I–II was measured (negative control). As a control for positive interaction, *hs*PURA I–III was used. While for *hs*PURA I–III the signal intensities above the background level clearly indicated dimerization (*Figure 3F*), protein fragments with either the F233del or the R245P variant showed strongly reduced signal intensities. This observation implies impairment of PURA dimerization in patients bearing either of these heterozygous variants.

## N-terminal PUR domain shows great flexibility in its domain fold

For a better understanding of the structural impact of genetic variations in the N-terminal PUR domain (*hs*PURA I–II), we determined its structure (fragments E57–E212) by X-ray crystallography (*Figure 4A* and *Supplementary file 1b*). The N-terminal PUR domain resembles the previously published PURA structure from *D. melanogaster* (*Graebsch et al., 2009*; *Weber et al., 2016*) (*dm*PURA I–II WT) with an average r.m.s.d. of 1.2 Å (*Figure 4—figure supplement 1*). However, in contrast to this invertebrate structure, we observed for *hs*PURA I–II that the four molecules within its asymmetric unit of the crystal lattice showed marked differences in its loop regions and in particular at the edges of the β-sheets (*Figure 4B*). This suggested greater structural flexibility than average domain folds and most likely than its *D. melanogaster* ortholog.

## Effects of patient mutations in the N-terminal PUR domain on RNA binding

To assess the interaction of *hs*PURA I–II with nucleic acids and to understand how selected patient-derived variants affect this function, we performed EMSAs for this protein fragment (*Figure 4C, D*). While *hs*PURA I–II quantitatively bound a 24-mer CGG-repeat RNA, the *hs*PURA I–II K97E-variant protein showed moderately but significantly decreased RNA binding (*Figure 4C, D*). Impaired RNA binding upon K97E mutation is not surprising as a change of the surface electrostatic potential from positive to negative may impact its interaction with nucleic acids (*Figure 3—figure supplement 1B*). In contrast, *hs*PURA I–II R140P did not show impaired RNA binding (*Figure 4C, D*). This finding indicates that R140P might be a benign variant and does not have disease-causing properties. As a negative control, the N-terminal PUR domain containing 11 selected point mutations was employed (*hs*PURA I–II m11; *Figure 1—figure supplement 2*). Since these mutations were introduced to abolish the interaction of *hs*PURA with nucleic acids, this mutated protein showed no detectable RNA binding in EMSA.

## PURA-dependent strand separation of dsDNA is affected by disease-causing mutations

We recapitulated the previously reported unwindase activity (*Darbinian et al., 2001*; *Weber et al., 2016*) with human PURA I–II and asked if patient-derived variants in this domain impair PURA's strand-separating activity. In vitro unwinding experiments with dsDNA were performed for *hs*PURA I–II and compared to the variants *hs*PURA I–II K97E and *hs*PURA I–II R140P as well as the *hs*PURA I–II m11 variant. While *hs*PURA I–II showed considerable strand-separating activity in this assay, the nucleic acid-binding-deficient m11 variant expectedly failed to separate dsDNA (*Figure 4E, F*). With an effect considerably stronger than observed for RNA binding (*Figure 4C, D*), the *hs*PURA I–II K97E variant showed a loss of strand-separation activity by 90% (*Figure 4E, F*). In contrast, *hs*PURA I–II R140P variant showed wild-type-like separation of dsDNA (*Figure 4E, F*), further supporting our previous interpretation from RNA-binding analyses that the genetic variant R140P is likely functional.

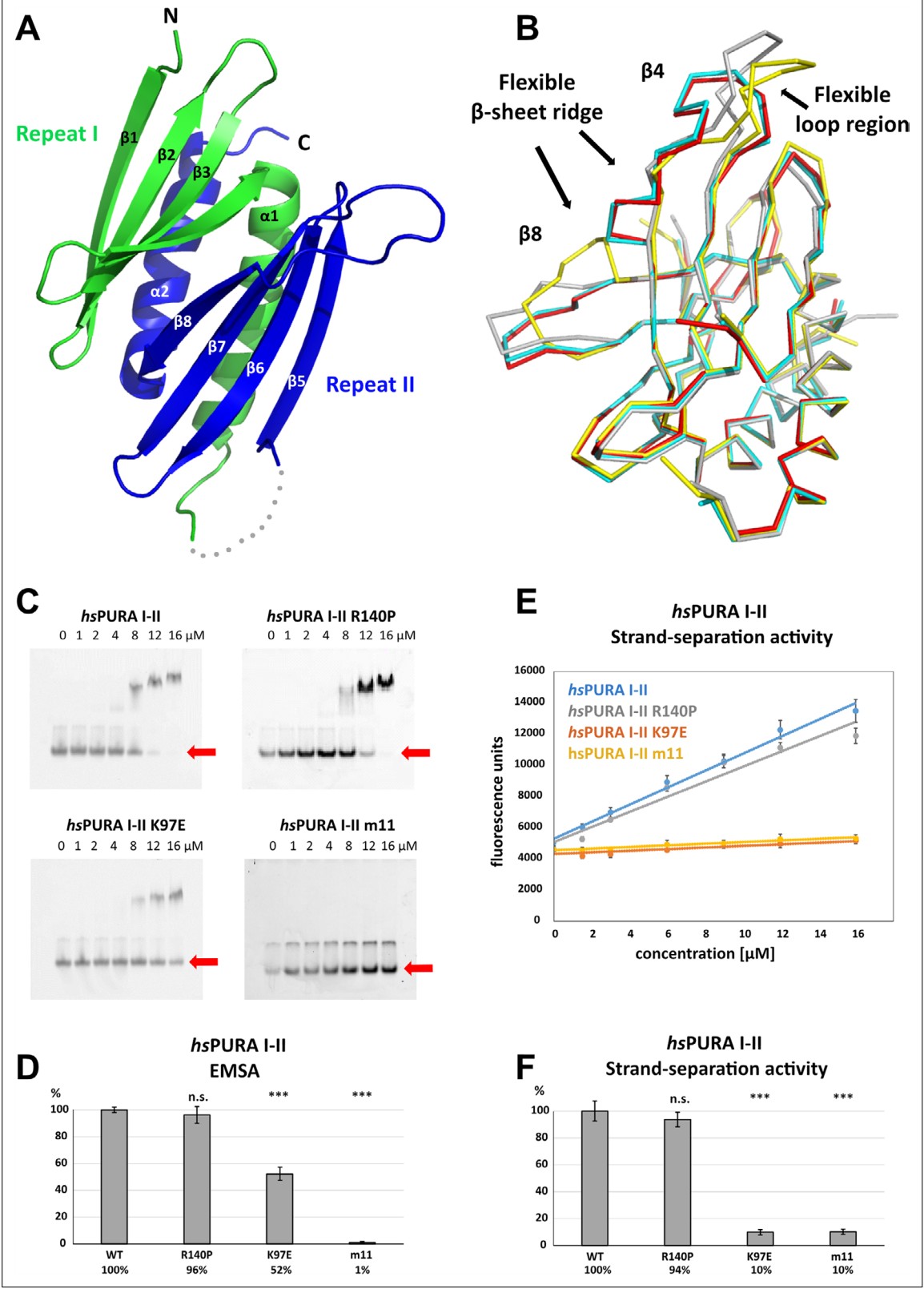

**Figure 4.** Structure and nucleic acids interaction of the N-terminal PUR domain. (**A**) Crystal structure of wild-type *hs*PURA repeats I (green) and II (blue) at 1.95 Å resolution. Protein fragment for which no electron density was visible is shown as a gray dotted line. (**B**) Ribbon presentation of the overlay of all four chains (shown in different colors) of *hs*PURA I–II in the asymmetric unit of the crystal. There are three regions showing greater differences in folding between these individual molecules, indicating that they are flexible and likely adopt several conformations also in solution. (**C**) In

*Figure 4 continued on next page*

*Figure 4 continued*

electrophoretic mobility shift assays (EMSAs), the interaction of different *hs*PURA I–II variants with 24-mer RNA (CGG)$_8$ labeled with Cy5-fluorophore was observed. The amount of labeled RNA was kept constant at 8 nM while the protein concentration increased from 0 to 16 µM. The unbound RNA was used for quantification (see D) and is indicated with the red arrow. Above the unbound RNA a second band of free RNA is visible, which might constitute a different conformation or RNA dimer. (**D**) Quantification of the RNA interactions of different *hs*PURA I–II variants from EMSAs shown in (**C**). The RNA-binding affinity of wild-type *hs*PURA was normalized to 100%. Pairwise *t*-test was used to assess differences in RNA binding compared to the *hs*PURA I–II. The variants K97E and m11 showed significantly lower relative affinities (p = 1.5E−05 and p = 4.6E−09, respectively). In contrast, the variant R140P did not alter binding affinity (p = 0.5). (**E**) Strand-separation activity of different *hs*PURA variants. For each *hs*PURA sample at least three measurements has been performed. The graphs show the averaged values as points as well as the standard deviations as error bars. The strand-separation activity shows linear increase within the used concentration range. (**F**) Quantitative representation of strand-separating activity of the *hs*PURA I–II variants. The strand-separating activity of *hs*PURA I–II was quantified from the slope in (**E**) and normalized to 100%. Except for R140P (p = 0.07), pairwise *t*-tests of the *hs*PURA I–II variants showed significantly lower relative activity (K97E p = 3.9E−16 and m11 p = 9.2E−16) than *hs*PURA I–II. For each experiment and each protein variant three replicates were measured, the standard deviations were calculated and are shown as bars. Asterisks in (**D**) and (**F**) indicate significance level: *** for p ≤ 0.001; n.s. for p > 0.05.

The online version of this article includes the following source data and figure supplement(s) for figure 4:

**Source data 1.** Electrophoretic mobility shift assays (EMSAs) for the interaction of *hs*PURA variants with a 24-mer (CGG)$_8$ RNA fluorescently labeled with Cy5 fluorophore at 5′-end.

**Source data 2.** Electrophoretic mobility shift assays (EMSAs) for the interaction of *hs*PURA variants with a 24-mer (CGG)$_8$ RNA fluorescently labeled with Cy5 fluorophore at 5′-end.

**Source data 3.** Electrophoretic mobility shift assays (EMSAs) for the interaction of *hs*PURA variants with a 24-mer (CGG)$_8$ RNA fluorescently labeled with Cy5 fluorophore at 5′-end.

**Source data 4.** Electrophoretic mobility shift assays (EMSAs) for the interaction of *hs*PURA variants with a 24-mer (CGG)$_8$ RNA fluorescently labeled with Cy5 fluorophore at 5′-end.

**Source data 5.** Electrophoretic mobility shift assays (EMSAs) for the interaction of *hs*PURA variants with a 24-mer (CGG)$_8$ RNA fluorescently labeled with Cy5 fluorophore at 5′-end.

**Source data 6.** Electrophoretic mobility shift assays (EMSAs) for the interaction of *hs*PURA variants with a 24-mer (CGG)$_8$ RNA fluorescently labeled with Cy5 fluorophore at 5′-end.

**Source data 7.** Electrophoretic mobility shift assays (EMSAs) for the interaction of *hs*PURA variants with a 24-mer (CGG)$_8$ RNA fluorescently labeled with Cy5 fluorophore at 5′-end.

**Source data 8.** Electrophoretic mobility shift assays (EMSAs) for the interaction of *hs*PURA variants with a 24-mer (CGG)$_8$ RNA fluorescently labeled with Cy5 fluorophore at 5′-end.

**Figure supplement 1.** Superposition of human and *Drosophila* PURA I–II structure.

## Circular dichroism spectroscopic measurements indicate impaired domain folding of the K97E variant

In order to confirm the structural integrity of the soluble *hs*PURA variants, we performed circular dichroism (CD) spectroscopy experiments. The spectra for both, *hs*PURA I–II and the *hs*PURA I–II R140P variant showed a typical alpha–beta profile (*Figure 5A*), suggesting a proper folding of the protein. In contrast, the spectrum for *hs*PURA I–II K97E indicated a strongly altered secondary structure content. This observation was unexpected as we had predicted that this surface mutation would not affect the protein's structural integrity this study and *Reijnders et al., 2018*. It also indicates that more detailed structural information of *hs*PURA I–II is needed to better understand its susceptibility to disease-causing variants of surface-exposed residues.

## *hs*PURA bearing mutation R140P with uncertain clinical significance shows wild-type-like domain folding

The genetic variation R140P had been previously described to be of uncertain clinical significance and it remained to be shown if t affects the protein's domain folding. CD spectra had already suggested that this mutation has no dramatic effect on the overall domain folding (*Figure 5A*). To assess if this variant has a rather local effect on the structural integrity of PURA that may not be detectable by CD spectroscopy, we solved the crystal structure of *hs*PURA I–II R140P at 2.15 Å resolution (*Figure 5—figure supplement 1A, B* and *Supplementary file 1b*). The crystals of R140P variant protein exhibited the same space group, unit cell parameters, and crystal packing as the wild-type protein (*Supplementary file 1b*). The R140P variant is located at the end of the flexible linker between PUR repeats I and

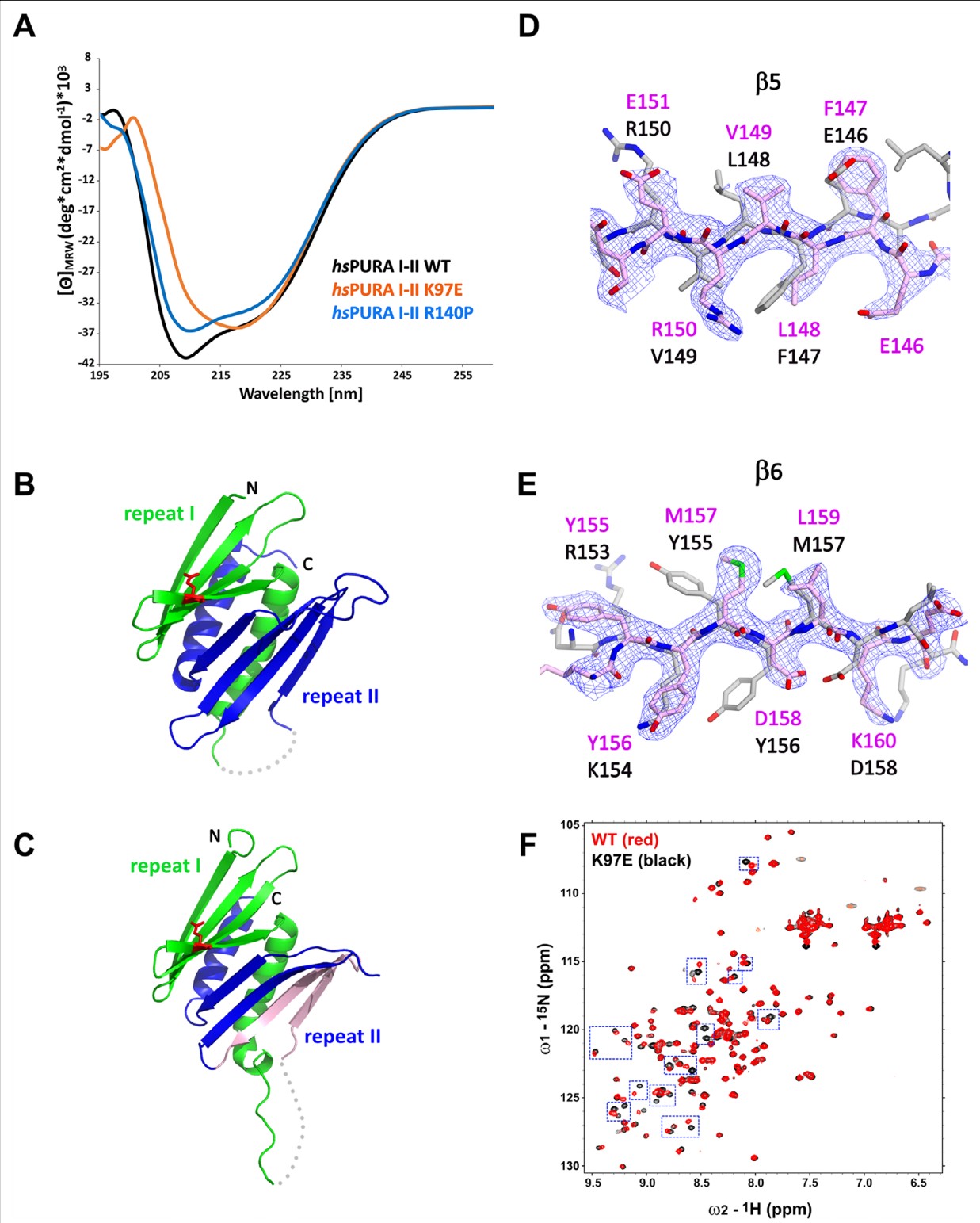

**Figure 5.** Structural analysis of *hs*PURA I–II bearing the K97E patient-derived variant. (**A**) Circular dichroism (CD) spectroscopic analyses of *hs*PURA I–II, K97E, and R140P variants. Shown are the mean of three CD measurements for each of the proteins (*n* = 3). The spectra of *hs*PURA I–II K97E show a very different profile, indicating an altered fold of this variant compared to the wild-type form. (**B, C**) Crystal structures of *hs*PURA K97E repeats I (green) and II (blue) at 2.45 Å resolution, showing a high overall similarity to *hs*PURA (*Figure 4A*). Two independent *hs*PURA I–II K97E chains (**A, B**) in the asymmetric unit are shown in (**B**) and (**C**), respectively. The mutated amino acid K97E is indicated as red sticks. Positional shifts of amino acids in β5 (**D**) and β6 (**E**) strands of the chain B in the crystal structure of *hs*PURA I–II K97E. 2Fo–Fc electron density map (contour 1σ) is shown for the selected fragment of β5

*Figure 5 continued on next page*

*Figure 5 continued*

and β6 strands in chain B (light pink). The superimposed chain A (gray) shows a register shift in chain B by +1 amino acid in the β5 strand and by +2 in the β6 strand. (**F**) Nuclear magnetic resonance (NMR) experiment with wild-type *hs*PURA I–II and *hs*PURA I–II K97E. Overlay of the $^1$H, $^{15}$N-HSQC spectra of *hs*PURA I–II (red) and K97E (black). Blue boxes indicate examples of changes between both spectra.

The online version of this article includes the following figure supplement(s) for figure 5:

**Figure supplement 1.** Structure of *hs*PURA I–II R140P variant.

**Figure supplement 2.** Comparison of structural features between wild-type and K97E-variant *hs*PURA protein.

**Figure supplement 3.** Comparison of structural features between wild-type and K97E-variant *hs*PURA protein.

II connecting helix α1 to the strand β5 (*Figure 1A*, *Figure 5—figure supplement 1A, B*). Because of its position in a flexible region, the mutated residue is partially visible only in one out of four PURA chains in the asymmetric unit. The overlap of the R140P-variant structure with the PURA I–II WT fragment confirmed our previous observation about the unusually high flexibility of the strands β4 and β8 forming the β-ridge (*Figure 5—figure supplement 1A, B*). Since no major structural differences were visible between *hs*PURA I–II WT and R140P variant (average r.m.s.d. of 0.89 Å for 133 Cα atoms aligned, *Figure 5—figure supplement 1C*), we concluded that this genetic variation is unlikely to impair the structural integrity of *hs*PURA.

## *hs*PURA I–II with the patient variant K97E reveals promiscuous domain folding

To understand how exactly the surface-exposed K97E mutation alters the domain folding of PURA (*Figure 5A*), we solved the crystal structure of *hs*PURA I–II K97E (*Figure 5B–E*, *Supplementary file 1b*). The asymmetric unit of the crystal contained two molecules of *hs*PURA I–II K97E and showed good overall fold similarity to the *hs*PURA I–II structure (*Figure 5B, C*). However, an overlay of both molecules of the K97E-variant protein from the asymmetric unit also revealed unusual differences between them with r.m.s.d. of 1.48 Å for 125 superimposed Cα atoms. While one of these molecules (chain A, *Figure 5B*) showed a fold very similar to *hs*PURA I–II (average r.m.s.d. 1.09 Å, *Figure 5—figure supplement 2A*), the second molecule (chain B) displayed much greater structural changes (average r.m.s.d. 1.61 Å, *Figure 5C* and *Figure 5—figure supplement 2A*). These differences were especially pronounced in PUR repeat II (*Figure 5D, E*), where we observed a shift of register in the β-sheet by one amino acid in strand β5 (*Figure 5D*) and by two amino acids in strand β6 (*Figure 5E*). More C-terminally, the loop linking β6 and β7 is shorter by two residues, thereby correcting again the sequence register within the next β-strands (β7 and β8).

Already the different molecules of wild-type *hs*PURA I–II in the crystal unit cell revealed considerable conformational freedom (*Figure 4B*, *Figure 5—figure supplement 2B, C*), not only for narrow movements of the loops but also in the folding of the β-strands. For instance, in some of the molecules in the asymmetric unit of the wild-type *hs*PURA I–II crystal, β4 and β8 strands formed bulges (*Figure 4B*, *Figure 5—figure supplement 2B, C*). From all the areas of the structure, the β-ridge appears to be structurally the most divergent part. It is surprising that the K97E point mutation, which is located on the β4 strand of the β-ridge, induces the observed unusual change in the β-sheet arrangement (*Figure 5C–E*) without destroying the domain's overall scaffold. Together these findings indicate great structural flexibility of the N-terminal PUR domain and promiscuity in folding, which results in misfolding upon mutation of a surface-exposed residue.

While CD measurements suggest a difference in folding between wild-type and K97E mutated PURA (*Figure 5A*) and the crystal structure of *hs*PURA I–II K97E indicates folding promiscuity (*Figure 5C–E*, *Figure 5—figure supplement 2D–F*), these data do not fully exclude unfolding of the domain in solution. To distinguish between a mutation-induced unfolding and an altered folded state, we performed nuclear magnetic resonance (NMR) measurements with the $^{15}$N-isotope-labeled samples of wild-type and K97E-variant *hs*PURA I–II (E57–E212 fragments). First, $^1$H,$^{15}$N-correlation spectra (HSQC) of both wild-type and K97E constructs showed a clear presence of the structured core, as evidenced by the well-dispersed resonances, with a significant number of residues adopting strand conformation ($^1$H 8.5–9.5 ppm) (*Figure 5F*). Next, we carefully compared the two HSQC spectra for individual resonances/residues. When the two spectra were overlaid, we observed that a significant number of resonances, especially in the region corresponding to the β-strand, were perturbed, which is clearly more

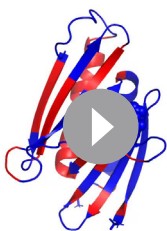

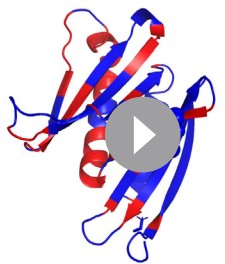

**Video 1.** Molecular dynamics simulations for the wild-type *hs*PURA I–II (chain B). Red color indicates the residues with significantly higher fluctuation described as p-value computed using the average position of the carbon alpha during the simulation as the reference point. 10 production runs were performed at the temperature of 310 K, resulting in 20 simulations of 100 ns each (1000 frames), or a total aggregate time of 1 µs for each protein. The systems were analyzed using root means square fluctuations (RMSF). More precisely, the RMSF was computed for each Cα atom in each simulation, leaving 10 RMSF data points per residue.
https://elifesciences.org/articles/93561/figures#video1

**Video 2.** Molecular dynamics simulations for the *hs*PURA I–II K97E (chain B). Red color indicates the residues with significantly higher fluctuation described as p-value computed using the average position of the carbon alpha during the simulation as the reference point. 10 production runs were performed at the temperature of 310 K, resulting in 20 simulations of 100 ns each (1000 frames), or a total aggregate time of 1 µs for each protein. The systems were analyzed using root means square fluctuations (RMSF). More precisely, the RMSF was computed for each Cα atom in each simulation, leaving 10 RMSF data points per residue.
https://elifesciences.org/articles/93561/figures#video2

extensive than just affecting the residues adjacent to the mutational site of residue 97 (*Figure 5F*). Finally, we performed the $^{15}$N-heteronuclear NOE (Nuclear Overhauser Effect) experiment on the K97E variant to measure the residue-level backbone dynamics (*Figure 5—figure supplement 3A*). The overlaid spectra of the absence (control) and presence (NOE) of $^1$H-saturation show most resonances retaining their signal intensities (rigid, hetNOE ~0.8), while the resonances corresponding to the dynamic regions, likely N-/C-terminal and loop regions, with significantly reduced signal intensities or even negative (flexible). Importantly, all the resonances perturbed upon K97E mutation remained in the presence of $^1$H-saturation, confirming that the structural rearrangement does not induce protein unfolding. Together, this shows that the K97E mutation does not induce global unfolding of *hs*PURA I–II, but rather triggers structural rearrangements mostly in the β-strand region.

To obtain a better understanding of the intrinsic dynamics of *hs*PURA I–II we performed in silico molecular dynamics simulations. The *hs*PURA I–II WT and K97E showed a similar overall profile (*Figure 5—figure supplement 3B*), where most of the flexibility or fluctuations are concentrated in the loops connecting strands β3 with β4 and β7 with β8. However, there are some statistically significant differences between the two proteins. The most significant differences concentrate on the two ends of the α1 helix, as well as the β5, β6, and β7 segments. In these regions, the K97E variant shows higher flexibility than the WT, with the N-terminal part of the α1 helix and the C-terminal end of β5 and β6 having the greatest gap between the two variants. The video for *hs*PURA I–II WT (*Video 1*) and for *hs*PURA I–II K97E (*Video 2*) of the simulation confirms a large degree of fluctuation in this area, which can adopt many different conformations in both WT and the mutated variant. Finally, there is one region where the WT displays a higher flexibility than the K97E variant, the loop connecting β7 with β8 (residues 175–185), although only a few of those differences are significant. Of note, while these simulations are consistent with the increased flexibility of the K97E variant already suggested by the crystal structure, the time window of these simulations was too short to observe alterations between the two folding states as seen in this experimental structure.

## Discussion

Currently, more than 270 different pathogenic variants in the *PURA* gene have been identified in over 650 patients with confirmed PURA syndrome (personal communication: PURA Foundation Australia). The aim of this study was to understand how genetic variations causing this neurodevelopmental

disorder affect the functional and structural integrity of the *hs*PURA protein. Toward this goal, we performed cellular, biophysical, and structural analysis with wild-type *hs*PURA as well as with versions of *hs*PURA bearing representative patient-derived variations.

To understand how genetic variations causing PUR syndrome affect the subcellular localization of *hs*PURA, we performed immunofluorescence microscopic analyses in HeLa cells by overexpressing full-length *hs*PURA as well as disease-causing variants in each of the three PUR repeats, that is K97E, I206F, and F233del. Consistent with previous reports (*Daigle et al., 2016*; *Markmiller et al., 2018*), we observed that *hs*PURA co-localizes to stress granules in HeLa cells (*Figure 1B, C*). As we did not observe significant changes in the association of patient-related variants of *hs*PURA to stress granules, it is suggested that this feature may not constitute a major hallmark of the PURA syndrome. It should be noted however that this interpretation must be considered with some caution as the experiments were performed in a PURA wild-type background.

What is more, the knockdown of PURA resulted in reduced sizes of those stress granules (*Figure 1—figure supplement 4*), indicating that PURA indeed does have an influence on these granules. Future experiments will be necessary to unambiguously clarify the importance of PURA's localization to stress granules for the etiology of PURA syndrome.

Very recently, *hs*PURA was reported to localize to P-bodies and that a PURA knockdown leads to a significant reduction in HeLa cells (*Molitor et al., 2023*). While we recapitulated co-localization of *hs*PURA to P-bodies (*Figure 2*), *hs*PURA K97E and F233del variants failed to fully localize to P-bodies. Since the F233del mutation impairs the dimerization of *hs*PURA (*Figure 3F*), this finding suggests the requirement of dimerization for efficient P-body association.

To address the question of what drives *hs*PURA accumulation in these phase-separated granules, we utilized a *hs*PURA version lacking RNA-binding activity (m11, *Figure 1—figure supplement 2*). This mutated protein failed to localize to stress granules or P-bodies, indicating that RNA binding is essential for both co-localization events. A potential RNA-dependent mechanism to recruit *hs*PURA into stress granules or P-bodies could be liquid-phase separation of *hs*PURA when bound to RNAs. In vitro experiments testing this potential property of *hs*PURA failed to yield phase-separated droplets. In summary, these findings indicate that RNA binding of *hs*PURA is important for its co-localization into stress granules and P-bodies, while we have no evidence that phase separation of *hs*PURA plays a role.

In a recent study, it was shown that a *hs*PURA knockdown in HeLa and in fibroblast cells resulted in down-regulation of the essential P-body proteins LSM14a and DDX6 and, most likely as a direct consequence, in strongly impaired P-body formation (*Molitor et al., 2023*). In light of this report, our finding that patient-derived variants show impaired P-body association of *hs*PURA (*Figure 2*) provide further evidence for a potential role of P-body localization of *hs*PURA for the etiology of PURA syndrome.

It has been reported that protein domains with PC4-like fold interact with RNA and DNA and are responsible for the separation of the double-stranded nucleic acids (*Darbinian et al., 2001*; *Weber et al., 2016*; reviewed in *Janowski and Niessing, 2020*). In this study, we show that both, the N- and C-terminal PUR domains of human PURA can separate double-stranded nucleic acids (*Figures 3D and 4E, F*) and to bind single-stranded nucleic acids (*Figures 3B, C and 4C, D*). We also tested patient-derived variants of *hs*PURA to find out whether they exhibit impaired interaction with nucleic acids. For the K97E variant, we observed in EMSAs a reduced affinity to RNA (*Figure 4C, D*). Similar behavior of the K97E variant was observed in the strand-separation assay where it showed an almost complete loss of activity (*Figure 4E, F*). The latter is of interest since previous experiments had demonstrated that strand separation contributes to the neuroprotective activity of PURA in a *Drosophila* FXTAS disease model (*Weber et al., 2016*). It is therefore also probable that loss of strand-separation activity contributes to cellular abnormalities in *hs*PURA patie nts harboring the K97E mutation.

To establish a structural framework for these observations, we solved the crystal structures of *hs*PURA III (*Figure 3A*) and of *hs*PURA I–II (*Figure 4A*). These human structures were overall very similar to previously published crystal structures from *dm*PURA (*Figure 3—figure supplement 1A*; *Figure 4—figure supplement 1*; *Graebsch et al., 2009*; *Weber et al., 2016*). Based on the crystal structures of *hs*PURA III (*Figure 3A*), we were able to predict folding defects in the C-terminal PUR domain (*hs*PUR III) that are caused by the patient-derived genetic variations F233del and R245P (*Figure 3—figure supplement 4*). Deletion of phenylalanine 233 within the β2-strand changes the orientation of all

subsequent side chains within the β-sheet and likely destroys the interaction network required for the assembly of this secondary structure. A mutation of arginine 245 into proline likely disrupts β-strand folding by introducing a kink. Since repeat III mediates homodimer formation through swapping of its α-helices, the dimerization of *hs*PURA is potentially affected by folding defects induced by either the F233del or the R245P mutation. Indeed, we could experimentally confirm by nanoBRET measurements in HEK293 cells that dimerization of *hs*PURA I–III F233del and *hs*PURA I–III R245 is impaired when compared to wild-type *hs*PURA (*Figure 3F*).

Determination of the crystal structure of *hs*PURA I–II revealed larger flexible regions than the previously reported invertebrate structures (*Graebsch et al., 2010*; *Graebsch et al., 2009*; *Weber et al., 2016*), with considerable differences in the four molecules of *hs*PURA I–II in the asymmetric unit of the crystals (*Figure 4B*, *Figure 5—figure supplement 2B, C*). These observations provide clear evidence that in particular strands β4 and β8 of the N-terminal PUR domain are dynamic and can adopt considerable local differences within the β-sheet. This flexibility already suggests structural promiscuity of PUR domains. It also indicates that they can at least partially compensate for structural perturbance by mutations, resulting in folded domains of genetically altered proteins. However, since these genetic variations were reported to cause PURA syndrome, such modest structural alterations must nevertheless result in functional impairment.

The interpretation of structural flexibility and folding promiscuity of PUR domains found experimental support from the crystal structure of its K97E variant, whose mutated residue has a solvent-exposed side chain (*Figure 5B–E*). The corresponding structure shows in the β5 strand a shift by one (*Figure 5D*) and in the β6 strand by two amino acids (*Figure 5E*) in one of the molecules of the asymmetric unit. Surprisingly, these structural distortions did not result in major unfolding events of this PUR domain. Furthermore, NMR measurements with the $^{15}$N- and $^{1}$H-labeled N-terminal PUR domain yielded many spectral differences between the wild-type and K97E variants in particular within their β-sheets (*Figure 5F* and *Figure 5—figure supplement 3A*). Despite these considerable structural changes, the NMR data also confirmed that the K97E-variant protein remains folded in solution and indicates that PUR domains may be able to partially compensate for mutations by folding promiscuity. These findings were further supported by molecular dynamics simulations, indicating high intrinsic motions of the N-terminal PUR domain (*Figure 5—figure supplement 3B*).

Instead of hot-spot regions for disease-causing mutations, as reported for other genetic disorders such as neurodevelopmental Rett syndrome (*Lombardi et al., 2015*), pathogenic variations in PURA syndrome patients are found along the entire protein sequence (*Johannesen et al., 2021*; *Reijnders et al., 2018*). Also, in contrast to disorders such as the Rett syndrome, most of these variations cause the full spectrum of disease symptoms. Our findings suggest that the flexibility and promiscuity of *hs*PURA domain folding render this protein particularly susceptible to mutations, even when amino-acid substitutions impose only moderate changes to the local charge and stereochemistry. This property of PUR domains might offer a structural explanation for why so many genetic variations across the sequence of *hs*PURA result in the full disease spectrum.

Not in all cases is the pathogenic effect of a given genetic alteration as clear as in the case of K97E. For instance, the replacement of arginine 140 by proline in *hs*PURA was initially reported to cause PURA syndrome (*Lee et al., 2018*). Even though the corresponding patient did show symptoms, the severity and type of symptoms raised doubts about this initial diagnosis. In our assessment, neither the functional in vitro experiments nor the structural assessment of *hs*PURA I–II R140P yielded any significant differences from the wild-type protein (*Figure 4C–F* and *Figure 5—figure supplement 1*). These experimental findings are consistent with R140P being rather a benign variant. Furthermore, after the initial description of R140P to be causative for PURA syndrome (*Lee et al., 2018*), subsequent genomic assessment of a relative of this patient uncovered that this person had the same genetic variation without any clear symptoms. Consequently, the R140P variant is now listed in the NCBI ClinVar database to be of 'uncertain significance' (https://www.ncbi.nlm.nih.gov/clinvar/variation/192343/). Our data clearly suggest that this genetic variant is rather benign. In summary, an assessment of the different genetic variations in *hs*PURA indicates a clear correlation between defects observed in in vitro analyses and the reported pathogenicity in patients. Hence, in future such in vitro analyses may serve as a diagnostic tool to distinguish pathogenic from benign genetic variations.

Recently, two publications and one manuscript uploaded to a preprint server resorted to in silico predicted, AI-based structures to model the potential impact of PURA syndrome mutations on the

structural integrity of *hs*PURA. Instead of assessing the physiologically correct dimer consisting of two PUR repeats, all three studies analyzed in silico-predicted *hs*PURA half-structure consisting of a single PUR repeat III with incomplete dimerization domain (*Dai et al., 2023*; *López-Rivera et al., 2022*, *Colombo et al., 2023*). Such half PURA structures are very unlikely to exist in solution and hence predictions based on them are of very limited value or worse, prone to yield wrong interpretations. We expect that our experimentally determined structures of the human N- and C-terminal PUR domains offer more physiological structural reference points for future assessments of novel *hs*PURA mutations and their potential pathogenicity in patients.

In summary, this study provides the first functional and experimental structural assessment of the effects of PURA syndrome-causing variants on folding, nucleic acid binding, and unwinding, as well as on subcellular localization to stress granules and P-bodies. Some of the selected patient variants of *hs*PURA were previously predicted to have folding defects (*Reijnders et al., 2018*). Of note, for all these cases expression tests with recombinant proteins resulted in insoluble samples, indicating that these variant proteins indeed have impaired folding (*Supplementary file 1a*). Whereas predictions of such misfolding events appear reliable, the examples of R140P (benign) and K97E (pathogenic) indicate that structural predictions of surface-exposed residues seem more complicated. In fact, the observed great promiscuity of folding of PUR domains indicates that also moderate surface mutations such as K97E can result in promiscuous folding and the full spectrum of disease symptoms. Our study may serve as proof-of-principle that in vitro and structural studies are suitable tools to classify genetic variants in the *hs*PURA gene as potentially pathogenic or benign.

# Materials and methods

## Establishment of stable cell lines

HeLa Kyoto cells were grown in DMEM (Dulbecco's Modified Eagle Medium; 4.5 g/l glucose, L-glutamine, phenol red; Invitrogen), supplemented with 10% fetal bovine serum (FBS), and 100 U/ml penicillin/streptomycin. Stable cell lines were established using the PiggyBac transposon system and transfected with Lipofectamine 3000. After 3 days of incubation, the medium was exchanged and supplemented with 700 µg/ml Hygromycin B (Invitrogen) for selection over 3 weeks. Stable cell lines were sorted for GFP signal by flow cytometry for similar expression levels at the core facility Flow Cytometry of the Biomedical Centre Munich (Munich University). Expression levels were confirmed by western blotting.

## Stress treatment

For each cell line, protein expression in $1 \times 10^5$ cells was induced by doxycycline at a final concentration of 1 µg/ml the day before stress treatment. Cells were treated for 1 hr with 500 µM sodium arsenite to induce stress-granule formation. As a reference, untreated cells received only media. After stress induction, stress media was removed, and cells were analyzed by immunofluorescence staining.

## Immunofluorescence staining

Cells grown on NO 1.5 coverslips were washed once with phosphate-buffered saline (PBS), fixed with 3.7% formaldehyde (diluted in PBS) for 10 min, and washed twice with PBS. Then, cells were incubated for 10 min with blocking buffer (1% donkey serum diluted in PBST (phosphate buffered saline with Tween 20)), before adding the primary antibody (diluted in blocking buffer) for 1 hr at room temperature. Subsequently, cells were washed three times for 5 min with PBST and the secondary antibody (diluted in blocking buffer) was added for 1 hr at RT in the dark. Afterward, cells were washed three times with PBST before DAPI (4',6-diamidino-2-phenylindole) (0.5 µg/ml) staining and mounting of the coverslips in Prolong Diamond Antifade Mountant onto the microscope slide.

## Microscopy

Confocal microscopy was performed at the Biomedical Center (Core facility Bioimaging; Munich University) with an inverted Leica SP8 microscope, equipped with lasers for 405, 488, 552, and 638 nm excitation. Either a 100× 1.4 or a 63× 1.4 oil objective was used to acquire the images (pixel size: 80 nm) using the fluorescence settings DAPI: 415–470 nm, GFP: 498–535, Cy3: 562–620, Cy5:

648–710. GFP, Cy3, and Cy5 were recorded with hybrid photodetectors (HyDs), DAPI with a conventional photomultiplier tube.

## Quantitative analysis of microscopic images with ImageJ

Images were processed using the Fiji:ImageJ software, applying only linear enhancements for brightness and contrast. Within one experiment, equal exposure times and processing conditions were used for all samples in respective channels. Co-localization of *hs*PURA with DCP1A and G3BP1 was quantified using the magic wand tool of Fiji:ImageJ software to determine the intensities ($I_{granule}$) of the processing bodies (PBs) or stress granules. Then, a margin of 0.5 µm for PBs and 0.3 µm for stress granules was drawn around the granule structure and intensities measured ($I_{band}$). Background intensity ($I_{background}$) was determined at a spot outside of cells and finally, the ratio of $I = \frac{I_{granule} - I_{background}}{I_{band} - I_{background}}$ was calculated.

## siRNA-mediated *PURA* knockdown in HeLa cells

The siRNA-mediated PURA knockdown was performed as described previously (*Molitor et al., 2023*). Briefly, approximately $1 \times 10^5$ HeLa cells were plated in 24-well plates. The next day, when the cells reached around 30% confluency, lipofection was performed using RNAiMAX (Thermo Fisher Scientific) according to the manufacturer's instruction. For that, a pre-designed PURA siRNA pool was used (Dharmacon, M-012136-01-000) and as a control siGENOME Non-Targeting Pool #1 (Dharmacon, D-001206-13-05). 20 µm of siRNA, either PURA siRNA or Control siRNA, was mixed with OptiMEM (Thermo Fisher Scientific) in one reaction tube, and lipofectamine RNAiMAX was mixed with OptiMEM in another. After an incubation time of 5–10 min at room temperature, both reactions were mixed and further incubated for 20 min at room temperature. Subsequently, the culture media of HeLa cells was changed to 250 µl of FMEM (F-12 Nutrient Mixture) +10% FBS + 1% penicillin/strepatvidin. Finally, 100 µl of the lipofection mixture was added drop-wise on each 24-well of HeLa cells. The transfected cells were incubated for 72 hr at 37°C and 5% $CO_2$ before being further used for immunofluorescence analysis.

## Quantitative analysis of immunofluorescence staining in *PURA*-knockdown experiments

Quantification of stress granules in PURA wild-type versus PURA knockdown HeLa cells was done using Fiji software (version 2.3.0/1.53q) (*Molitor et al., 2023*). Zeiss files (czi) were opened in Fiji and the Channels were split in DAPI and G3BP1 channels. Then, the cell number was determined by using the auto threshold function on the DAPI channel and analyzing particles (Analyze particle function) with a particle size of 1000-infinity pixel units, which should cover all DAPI signals. G3BP1 granules were quantified using the same functions but selecting a particle size of ≥0.7 µm² and a circularity of 0.1–1.00. Size comparison was done by comparing the average size per granule within the particle size analysis function. Quantification of stress granules was done in biological triplicates and technical triplicates with a magnification of ×20.

## Purification of total HeLa RNA

RNA from $1 \times 10^7$ HeLa cells was extracted using the TRIzol Plus RNA Purification Kit (Thermo Fisher Scientific) according to the manufacturer's instructions and following the general precautions required for RNA work. RNA was eluted in 100 µl diethyl pyrocarbonate-treated water. Integrity, purity, and amount of total HeLa RNA were determined using Agilent Tape station measurements according to the manufacturer's instructions.

## In vitro phase separation assays

Full-length wild-type *hs*PURA (aa 1–322) and FUS RGG3-PY (aa 454–526) were thawed and FUS RGG3-PY was incubated at 95°C for 5 min and subsequently kept at room temperature. Proteins were diluted in condensate buffer (20 mM $Na_2HPO_4/NaH_2PO_4$, pH 7.5, 150 mM NaCl, 2.5% glycerol, 1 mM DTT (dithiothreitol)), and total HeLa cell RNA was added at an RNA-to-protein mass ratio of 0.5. For visualization of condensates, samples were immediately transferred to self-assembled sample chambers formed by double-sided sticky tape, taped onto a glass slide, and sealed with a coverslip.

Condensates were imaged after 15 min using phase contrast microscopy and a HC PL Fluotar L ×40/0.6 PH2 objective on a Leica DMi8 microscope (Leica, Germany).

## NanoBRET

HEK293 cells were seeded in 6-well plates at 800,000 cells per well with culture media (89% DMEM (Gibco); 10% FBS; 1% Anti-Anti (Gibco)) and incubated for 6 hr at 37°C/5% $CO_2$. Cells were transfected using Lipofectamine 3000 (Thermo Fisher Scientific) and a ratio of 1:100 for HaloPURAI-III and NLucPURAI-III mutants. Cells were incubated for 24 hr at 37°C/5% $CO_2$, trypsinized, adjusted to a final density of 200,000 cells/ml with assay medium (95% Opti-MEM I Reduced Serum Medium; 4% FBS; 1% Anti-Anti), and divided into two pools. Next, either HaloTagNanoBRET 618 Ligand or dimethyl sulfoxide as no-ligand control was added to each pool at concentrations of 0.1 M and 0.1%, respectively. Three technical replicates of each cell suspension (20,000 cells) were transferred in a white Lumitrac 96-well plate (F-Bottom) and incubated for 18 hr at 37°C/5% $CO_2$. 25 µl diluted Nano-BRET Nano-Glo Substrate (1:100 in Opti-MEM I Reduced Serum Medium, no phenol red) was added to all wells and mixed for 30 s. Plates were measured using a Tecan Spark plate reader with the following settings: HaloTagNanoBRET 618 Ligand: 595–650 nm (bandwidth 27.5 nm); NanoLuciferase: 430–455 nm (bandwidth 12.5 nm); integration time: 1000 ms. Samples were provided as biological triplicates, of which each was measured as technical duplicates. As positive control the homotypic dimerization of wild-type *hs*PURA I–III was applied as a single sample on each 96-well plate. The experimental mBUs (milliBRET units) were calculated by (618 EM/460 EM) × 1000 × BU = mBU. From these mBU, the no-ligand control was subtracted.

Cloning, expression, and purification of *hs*PURA repeat I–II *hs*PURA protein fragments E57–E212 (PUR repeats I and II) was amplified by polymerase chain reaction (PCR) and cloned using in-fusion method (*Berrow et al., 2007*) into pOPINS3C vector and expressed in *E. coli* Rosetta cells using auto-induction media (*Studier, 2005*). The cell pellet was lysed in a buffer composed of 50 mM HEPES (4-(2-hydroxyethyl)-1-piperazineethanesulfonic acid) pH 7.5, 500 mM NaCl, 20 mM imidazole, supplemented with protease inhibitors (Roche) and DNAse, and centrifuged. The supernatant was applied on a 5 ml HisTrap column (GE Healthcare), washed with high-salt buffer (50 mM HEPES pH 7.5, 1 M NaCl, 20 mM imidazole), and eluted with a linear gradient from 20 to 400 mM imidazole. Protein was dialyzed against 50 mM HEPES 7.5, 200 mM NaCl, 1 mM DTT overnight at 4°C with PreScission protease added. After a subtractive HisTrap column, the flow-through was diluted in 50 mM HEPES pH 7.5, 70 mM NaCl and loaded onto a 1-ml HiTrap Heparin HP (GE Healthcare) column and eluted using a linear NaCl gradient (0.1–2 M) and subsequently applied onto a Superdex75 16/60 column (GE Healthcare) equilibrated with 20 mM HEPES pH 7.5 and 200 mM NaCl. The selected fractions were validated via sodium dodecyl sulfate–polyacrylamide gel electrophoresis (SDS–PAGE), pooled, and concentrated to 8 mg/ml.

Cloning, expression, and purification of *hs*PURA K97E repeat I–II *hs*PURA protein fragments E57–E212 with K97E mutation were cloned into pOPINJ expression vector as fusion protein with N-terminal His$_6$-GST-tag. After expression in *E. coli* Rosetta cells with LB media at 18°C and induction with 0.25 mM IPTG (isopropyl β-D-1-thiogalactopyranoside) at OD 0.6, cells were lysed by sonication in resuspension buffer (1 M NaCl, 50 mM HEPES, 2 mM DTT, pH 7.5, DNAse I, EDTA (ethylenediaminetetraacetic acid) free protease inhibitor (Roche)) and centrifuged for 30 min at 20,000 × *g* at 4°C. The soluble protein fraction was bound to a GSTrap FF column (GE Healthcare) and washed with lysis buffer, high-salt buffer (2 M NaCl, 20 mM HEPES, 2 mM DTT, pH 7.5), and dialysis buffer (300 mM NaCl, 20 mM HEPES, 2 mM DTT, pH 7.5 at 4°C). Bound protein was eluted with buffer containing 500 mM NaCl, 20 mM HEPES, 25 mM glutathione, pH 7.5. Protein was dialyzed overnight in dialysis buffer together with PreScission protease (100 µg per 5 ml). After a subtractive GSTrap column, the flow-through was diluted in 50 mM HEPES pH 7.5, 70 mM NaCl and loaded onto a 5-ml HiTrap Q FF (GE Healthcare) column and eluted using a linear NaCl gradient (0.1–2 M). The same procedure was repeated using HiTrap Heparin HP (GE Healthcare) column. As the last step, size exclusion chromatography has been performed in buffer containing 300 mM NaCl, 20 mM HEPES, 2 mM DTT, pH 7.5 at 4°C on Superdex75 16/60 column (GE Healthcare). The selected fractions were validated via SDS–PAGE, pooled, and concentrated to 8.2 mg/ml.

Cloning, expression, and purification of *hs*PURA R140P repeat I–II *hs*PURA protein fragments E57–E212 with R140P mutation were cloned into pOPINJ expression vector as fusion protein with

N-terminal His$_6$-GST-tag and expressed in *E. coli* Rosetta cells using auto-induction media at 22°C. The purification has been performed as described for *hs*PURA repeat I–II K97E mutant. The selected fractions were validated via SDS–PAGE, pooled, and concentrated to 5.6 mg/ml.

## Cloning, expression, and purification of *hs*PURA repeat I–II m11 variant

*hs*PURA protein fragments E57–E212 with 11 mutations abolishing nucleic acid binding (K71A, N80A, K82G, F85A, K87A, K97A, R153A, N162A, R164G, F167A, and R169A) was cloned into pOPINJ expression vector. The purification was performed using the same protocol as for *hs*PURA repeat I–II K97E variant. The selected fractions were validated via SDS–PAGE, pooled, and concentrated to 5.2 mg/ml.

## Cloning, expression, and purification of full-length WT *hs*PURA and full-length *hs*PURA m11 variant

The full-length wild-type *hs*PURA as well as its m11 variant were cloned into pOPINJ vector. Expression and purification of both proteins were performed as described above for the K97E variant. The selected fractions were validated via SDS–PAGE, pooled, and concentrated to 3.4 mg/ml (wt) and 1.6 mg/ml (m11), respectively.

## Cloning, expression, and purification of *hs*PURA repeat III

Fragment P215-K280 of *hs*PURA (PUR repeat III) was amplified by PCR and cloned into pOPINS3C vector. Expression and purification were performed as described above for *hs*PURA repeat I–II except for the exclusion of the Heparin column step. The selected fractions were validated via SDS–PAGE, pooled, and concentrated to 7.5 mg/ml.

## Electrophoretic mobility shift assay

EMSAs were performed in a final reaction volume of 20 µl in EMSA buffer composed of 300 mM NaCl, 20 mM HEPES pH 8.0, 2 mM DTT, 4% (vol/vol) glycerol, and 3 mM MgCl$_2$. The final protein concentration ranged from 0 to 16 µM, with a constant final concentration of fluorescence-labeled RNA 8 nM and 10 µg/ml of competitor yeast tRNA. The 24-mer RNA fragment (CGG)$_8$ was labeled on its 5′-end with Cy5 fluorophore (Eurofins Genomics). After incubation of the reaction for 30 min at RT, 5 µl of the samples were loaded onto the 6% TBE (Tris-Borate-EDTA) polyacrylamide gel and run for 45 min at 100 V. The fluorescence measurements were performed using an Amersham Typhoon scanner and software v 2.0.0.6. The intensities of the bands observed on the scanned EMSAs gels were quantified with FUJI ImageJ software (v 1.53C).

## Unwinding assay

For the fluorescence-based unwinding experiment the double-labeled DNA fragment, FAM-**CCAGG**GCACTTAAAAAAATTCG**CCTGG**-Dabcyl, was used (double-stranded part of the DNA is shown as bold). The assay was performed in 50 µl reaction volume in the unwinding buffer composed of 300 mM NaCl, 20 mM HEPES pH 8.0, 2 mM DTT, 4% (vol/vol) glycerol, and 3 mM MgCl$_2$. The protein concentration ranged from 0 to 16 µM, with a constant DNA concentration of 50 nM. The fluorescence measurements were performed using a Perkin Elmer EnVision 2104 Multilable Reader and Wallac EnVision Manager software v 1.12. FAM fluorescence was excited at 495 nm with a slit of 2 nm. Emission was recorded at 517 nm with a slit of 3 nm for 0.5 s (integration time). Experiments were performed at least three times.

## Circular dichroism

For secondary structure analysis of *hs*PURA I–II wt, K97E, and R140P, CD spectra were recorded using a Jasco J-715 spectropolarimeter (JASCO) and the range of 190–260 nm at 20°C. The measurements were performed with a 1-mm path length high precision quartz cuvette (Hellma Analytics). The concentration of all tested *hs*PURA variants was 20 µM in a buffer containing 60 mM NaCl, 20 mM HEPES, and 2 mM DTT at pH 7.5. Further measurement parameters included a scanning speed of 50 nm/min, three scans, and a response time of 8 s. Sensitivity was set to standard (100 mdeg).

## Crystallization

The crystallization experiments for *hs*PURA I–II, *hs*PURA I–II K97E, *hs*PURA I–II R140P, and *hs*PURA III were performed at the X-ray Crystallography Platform at Helmholtz Center Munich. For *hs*PURA I–II,

the best diffracting crystals were obtained from 0.1 M Tris buffer pH 8.2, 0.2 M sodium acetate, and 30% (wt/vol) PEG 4000. The best crystals for *hs*PURA I–II K97E grew in 0.1 M citric acid pH 4.0, 1 M LiCl, 9% (wt/vol) PEG 6000. *hs*PURA I–II R140P crystallized directly during protein concentration in the size exclusion buffer. For *hs*PURA III crystals were grown in 0.1 M Tris pH 8.5, 12% (vol/vol) glycerol, 1.5 M $(NH_4)_2SO_4$. For the X-ray diffraction experiments, crystals were mounted in the nylon fiber loops and flash-cooled to 100 K in liquid nitrogen. The cryoprotection was performed for a few seconds in reservoir solution complemented with 25% (vol/vol) ethylene glycol in all cases. Diffraction data for *hs*PURA I–II were collected on the PETRAIII P11 beamline, for *hs*PURA I–II K97E, *hs*PURA I–II R140P and *hs*PURA III the measurements were performed at Swiss Light Source, Paul Scherrer Institute, beamline X06DA. The best dataset for each protein variant was indexed and integrated using *XDS* (*Kabsch, 2010*) and scaled using *SCALA* (*Evans, 2006*; *Winn et al., 2011*). Intensities were converted to structure-factor amplitudes using the program *TRUNCATE* (*French and Wilson, 1978*). *Supplementary file 1a* summarizes data collection, processing, and refinement statistics.

## Structure determination and refinement

The structures of *hs*PURA I–II and *hs*PURA III were solved by molecular replacement using the crystal structures of *Drosophila* PURA repeats I–II (PDB ID: 3k44) (*Graebsch et al., 2009*) and repeat III (PDB-ID: 5FGO) (*Weber et al., 2016*) as search model, respectively. The structures of *hs*PURA I–II K97E and *hs*PURA I–II R140P were solved by molecular replacement using the structure of *hs*PURA I–II from this study as search model. In all cases, for the molecular replacement calculations the program *MolRep* (*CCP4*) was used. Model rebuilding in all cases was performed with *COOT* (*Emsley and Cowtan, 2004*). The refinement was done with *REFMAC5* (*Murshudov et al., 1997*) using the maximum-likelihood target function. The final model is characterized by $R$ and $R_{free}$ factors of 17.4/23.1%, 20.2/27.2%, 20.9/28.2%, and 17.6/22.4% for *hs*PURA I–II, *hs*PURA I–II K97E, *hs*PURA I–II R140P, and *hs*PURA III, respectively. The stereochemical analysis of the final model was done with *PROCHECK* (*Laskowski et al., 1993*) and *MolProbity* (*Chen et al., 2010*). For all the crystallographic calculations, the *SBGrid* software bundle was used (*Morin et al., 2013*). Atomic coordinates and structure factors have been deposited in the Protein Data Bank under the accession codes 8CHT (*hs*PURA I–II), 8CHU (*hs*PURA I–II K97E), 8CHV (*hs*PURA I–II R140P), and 8CHW (*hs*PURA III).

## Expression of isotope-labeled proteins for NMR

$^{15}$N-labeled proteins (*hs*PURA I–II wt and K97E mutant) were expressed in $^{15}$N-M9 minimal medium (1× $^{15}$N-labeled M9 salt solution, 0.4% glucose, 1 mM $MgSO_4$, 0.3 mM $CaCl_2$, 1 µg/l biotin, 1 µg/l thiamine, 1× trace metals) with respective antibiotics. 150 ml of pre-culture was grown overnight at 37°C and used for inoculation of 3 l pre-warmed M9 minimal medium the next day. After growing to $OD_{600\,nm}$ = 0.6 at 37°C, cells were induced with 0.25 mM IPTG, and proteins expressed at 18°C overnight (O/N) (approx. 18 hr).

## NMR

NMR measurements for HSQC comparison were performed with the $^{15}$N-labeled samples in buffer containing 300 mM NaCl, 20 mM HEPES, 0.2 mM TCEP (Tris(2-carboxyethyl)phosphine), and pH 7.5. Uniformly $^{15}$N-labeled NMR samples were prepared at protein concentrations of 25 µM (WT and K97E mutant) and 454 µM (K97E) in a buffer containing 300 mM NaCl, 20 mM HEPES, 0.2 mM TCEP, and pH 7.5 with 10% $D_2O$ for lock signal. NMR experiments were recorded at 298 K on 800- and 600-MHz Bruker Avance NMR spectrometers, equipped with cryogenic or room-temperature triple resonance gradient probes. NMR spectra were processed by *TOPSPIN3.5* (Bruker), then analyzed using *NMRFAM-SPARKY* (*Lee et al., 2015*). $^1$H-$^{15}$N heteronuclear NOE experiments were recorded on a 600-MHz spectrometer at 298 K in an interleaved manner with and without proton saturation.

## Molecular dynamics simulations

Molecular dynamics simulations were performed using the *ACEMD* engine (*Harvey et al., 2009*). The system was prepared for simulation using the HTMD (High-Throughput Molecular Dynamics) library (*Doerr and De Fabritiis, 2014*). Both, the *hs*PURA I–II WT and the K97E variant, were prepared starting from the available structures (PDB ID: 7PO8 – WT, chain B; and PDB ID: 7PRF – K97E mutant, chain B). Due to lack of resolved structure of the loop between α1 and β5 (the electron density map

was not visible) for the molecular dynamics calculations the missing parts were manually adjusted and linked. This part of the structure was not taken for the final analysis. The proteins were titrated using HTMD Protein Prepare, adding hydrogens and generating ionization states for the side chains using PROPKA at pH 7.0, and end-capping of N- and C-termini, using ACE (acetyl group) and NME (N-methyl group) capping. The prepared proteins were parameterized using the Amber 14 SB force field (*Wang et al., 2004*). The systems were solvated in a box of dimensions 70.6, 70.6, 70.6 Å for the K97E and 67.1, 67.1, 67.1 for WT; the chosen solvent model was TIP3P. The solvated systems were equilibrated for 3 ns using HTMD (High-Throughput Molecular Dynamics) equilibration protocols applying restraints on protein backbones for minimization. Equilibration was run at 310 K with NPT conditions (set of conditions used to control the temperature (T), pressure (P), and number of particles (N) in the simulation system) setting the cut-off for long-range PME (Particle Mesh Ewald) forces to 9 Å; a time step of 4 fs was used for the integration and energy calculation. 10 production runs were performed on the resulting equilibrated system at the temperature of 310 K for WT and K97E, resulting in 20 simulations of 100 ns each (1000 frames), or a total aggregate time of 1 µs for each protein. The systems were analyzed using root means square fluctuations (RMSF). More precisely, the RMSF was computed for each Cα atom in each simulation, leaving 10 RMSF data points per residue. The mean and standard deviation of these data points were calculated, and a two-sided *t*-Student test was performed on each pair of comparable residues, that is, residues that are paired in the sequence alignment of the two proteins.

## Acknowledgements

We acknowledge the use of the X-ray Crystallography Platform at Helmholtz Center Munich, the Core Facility Bioimaging of the BMC of the Ludwig-Maximilians University Munich, as well as the Bavarian NMR Centre (BNMRZ). We would like to thank Vera Roman for her excellent technical support as well as Melinda Anderson from the PURA Foundation Australia and the PURA Syndrome Foundation for their precious support.

## Additional information

### Funding

| Funder | Grant reference number | Author |
|---|---|---|
| Deutsche Forschungsgemeinschaft | Ni-1110/6-2 | Dierk Niessing |

The funders had no role in study design, data collection, and interpretation, or the decision to submit the work for publication.

### Author contributions

Marcel Proske, Sabrina Bacher, Data curation, Formal analysis, Validation, Investigation, Visualization, Writing - original draft, Writing - review and editing; Robert Janowski, Data curation, Formal analysis, Supervision, Validation, Investigation, Visualization, Writing - original draft, Writing - review and editing; Hyun-Seo Kang, Alejandro Varela-Rial, Data curation, Formal analysis, Validation, Investigation, Visualization, Writing - review and editing; Thomas Monecke, Data curation, Formal analysis, Supervision, Validation, Investigation, Visualization, Writing - review and editing; Tony Koehler, Roberto Fino, Data curation, Formal analysis, Validation, Investigation, Visualization; Saskia Hutten, Gianni De Fabritiis, Supervision; Jana Tretter, Anna Crois, Lena Molitor, Data curation, Formal analysis, Validation, Investigation; Elisa Donati, Data curation, Formal analysis, Supervision, Validation, Investigation, Writing - review and editing; Dorothee Dormann, Michael Sattler, Supervision, Writing - review and editing; Dierk Niessing, Conceptualization, Supervision, Funding acquisition, Validation, Writing - original draft, Project administration, Writing - review and editing

### Author ORCIDs

Robert Janowski ⬤ http://orcid.org/0000-0002-9940-2143
Thomas Monecke ⬤ http://orcid.org/0000-0003-3748-710X

Anna Crois [ORCID] http://orcid.org/0009-0001-0791-5719
Dierk Niessing [ORCID] http://orcid.org/0000-0002-5589-369X

Joint Public Review: https://doi.org/10.7554/eLife.93561.3.sa1
Author response https://doi.org/10.7554/eLife.93561.3.sa2

## Additional files

### Supplementary files

• Supplementary file 1. Summary of the experiments performed for *hs*PURA. (A) Summary of the experiments in this study for the selected human PURA variants. (B) Data collection and refinement statistics for crystal structures of human PURA variants. Values in parentheses are for the highest resolution shell.

• MDAR checklist

### Data availability

All data are available in the main text or the supplementary materials. Structural models and diffraction data are available at the Protein Data Base (PDB; https://www.rcsb.org/) under the accession codes 8CHT (hsPURA I-–II), 8CHU (hsPURA I-–II K97E), 8CHV (hsPURA I-–II R140P), and 8CHW (hsPURA III).

The following datasets were generated:

| Author(s) | Year | Dataset title | Dataset URL | Database and Identifier |
|---|---|---|---|---|
| Janowski R, Niessing D | 2024 | Crystal structure of human PURA (fragment Glu57-Glu212, PUR repeat I and II) | https://www.rcsb.org/structure/8CHT | RCSB Protein Data Bank, 8CHT |
| Janowski R, Niessing D | 2024 | Crystal structure of human PURA repeat I-II K97E mutant | https://www.rcsb.org/structure/8CHU | RCSB Protein Data Bank, 8CHU |
| Janowski R, Niessing D | 2024 | Crystal structure of human PURA (fragment Glu57-Glu212, PUR repeat I and II) R140P mutant | https://www.rcsb.org/structure/8CHV | RCSB Protein Data Bank, 8CHV |
| Janowski R, Niessing D | 2024 | Crystal structure of human PURA (fragment Pro216-Lys280, PUR repeat III) | https://www.rcsb.org/structure/8CHW | RCSB Protein Data Bank, 8CHW |

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
