## [Editor Report · eLife assessment]

This **important** study addresses the mechanisms by which mutations in the PURA protein, a regulator of gene transcription and mRNA transport and translation, cause the neurodevelopmental PURA syndrome. Based on **convincing** evidence from structural biology, molecular dynamics simulation, biochemical, and cell biological analyses, the authors show that the PURA structure is very dynamic, rendering it generally sensitive to structure-altering mutations that affect its folding, DNA-unwinding activity, RNA binding, dimerization, and partitioning into processing bodies. These findings are of substantial importance to cell biology, neurogenetics, and neurology alike, because they provide first insights into how very diverse PURA mutations can cause similar and penetrant molecular, cellular, and clinical defects.

---

## [Referee Report · Joint Public Review]

The present study focuses on the structure and function of human PURA, a regulator of gene transcription and mRNA transport and translation whose mutation causes the neurodevelopmental PURA syndrome, characterized by developmental delay, intellectual disability, hypotonia, epileptic seizures, a.o. deficits. The authors combined structural biology, molecular dynamics simulation, and various cell biological assays to study the effects of disease-causing PURA mutations on protein structure and function. The corresponding data reveal a highly dynamic PURA structure and show that disease-related mutations in PURA cause complex defects in folding, DNA-unwinding activity, RNA binding, dimerization, and partitioning into processing bodies. These findings provide first insights into how very diverse PURA mutations can cause penetrant molecular, cellular, and clinical defects. This will be of substantial interest to cell biologists, neurogeneticists, and neurologists alike.

A particular strength of the present study is the structural characterization of human PURA, which is a challenging target for structural biology approaches. The molecular dynamics simulations are state-of-the-art, allowing a statistically meaningful assessment of the differences between wild-type and mutant proteins. The functional consequences of PURA mutations at the cellular level are fascinating, particularly the differential compartmentalization of wild-type and mutant PURA variants into certain subcellular condensates.

---

## [Author Response]

The following is the authors’ response to the original reviews.

**Public Review**
[...] A particular strength of the present study is the structural characterization of human PURA, which is a challenging target for structural biology approaches. The molecular dynamics simulations are state-of-the-art, allowing a statistically meaningful assessment of the differences between wild-type and mutant proteins. The functional consequences of PURA mutations at the cellular level are fascinating, particularly the differential compartmentalization of wild-type and mutant PURA variants into certain subcellular condensates.Weaknesses that warrant rectification relate to(i) The interpretation of statistically non-significant effects seen in the molecular dynamic simulations.

We removed from the manuscript the sentence which indicated that we analyzed statistically non-significant effects. Therefore, the above statement has been resolved.

(ii) The statistical analysis of the differential compartmentalization of PURA variants into processing bodies vs. stress granules, and

We re-analyzed all cell-biological data and adjusted the statistical analysis of P-bodies and Stress-granule intensity analysis. The new, and improved statistics have replaced the original analyses in the corresponding figures (Figs. 1C and 2B).

(iii) Insufficient documentation of protein expression levels and knock-down efficiencies.

Quantification of protein expression levels by Western blotting is shown in Appendix Figure S1. Quantification of knock-down efficiencies by Western blot experiments (Appendix Figure S3).

**Recommendations for the authors: Reviewer #1**
Concerns and Suggested Changes(a) I have only one concern about the computational part and that is about statements such as "There are also large differences in the residue surrounding the mutation spot (residues 90 to 100), where the K97E mutant also shows much greater fluctuation. However, these differences are not significant due to the large standard deviations." If the differences are not statistically significant, then I would suggest either removing such a statement or increasing the statistics.

We agree with the Reviewer’s comment. We removed this sentence from the text.

**Recommendations for the authors: Reviewer #2**
General CommentsThis is a challenging structural target and the authors have made considerable efforts to determine the effect of several mutations on the structure and function. Many of the constructs, however, could not be expressed and/or purified in bacteria. However, it is not clear to what extent other expression systems (e.g. *Drosophila* or human) were considered and if this would have been beneficial.

We did not use other expression systems because the wild-type protein is well-behaved when expressed in *E. coli*. In case a mutant variant cannot be expressed or does not behave well in *E. coli*, this constitutes a clear indication that the respective mutation impairs the protein’s integrity. Thus, by using *E. coli *as a reference system for all the variants of PURA protein, we could assess the influence of the mutations on the structural integrity and solubility. Only for the variants that did not show impairment in E. coli expression, we continued to assess in more detail why they are nevertheless functionally impaired and cause PURA Syndrome.

Concerns and Suggested Changes(a) The schematic in Figure 3A would have been helpful for interpreting the mutations discussed in Figures 1 and 2. I would suggest moving it earlier in the text.

We changed the figure according to the Reviewer’s suggestion.

(b) I believe the RNA used for binding studies in Figures 3C and D was (CGG)8. Are the two "free" RNA bands a monomer and a dimer (duplex?)?

Although we do not know for certain, it is indeed likely that the two free RNA bands represent either different secondary structures of the free RNA or a duplex of two molecules. Of note, PURA binds to both “free” RNA bands, indicating that it either does not discriminate between them or melts double-stranded RNA in these EMSAs.

There also seems to be considerable cooperativity in the binding, so I wonder if a shorter RNA oligonucleotide might facilitate the measurement of Kds.

The length of the used RNA was selected based on the estimated elongated size of the full-length PURA and the presence of 3 PUR repeats. Assuming that one PUR repeat interacts with about 6-7 bases (data from the co-structure of *Drosophila* PURA with DNA; PDB-ID: 5FGP) and that full-length PURA forms a dimer consisting of three PUR repeats, the full-length protein in its extended form should cover a nucleic-acid stretch of about 24 bases.

Also, it is not clear how the affinities were measured particularly for hsPURA III since free band is never fully bound at the highest protein concentration.

It was not our goal to measure Kds for the interaction of PURA variants with RNA. The EMSA experiments were conducted to detect relative differences in the interaction between PURA variants and RNA. To estimate the differences, we measured total intensity of the bound (shifted) and unbound RNA. The intensities of the bands observed on the scanned EMSA gels were quantified with FUJI ImageJ software. We calculated the percentage of the shifted RNA and normalized it. hsPURA III fragment shows much lower affinity therefore it does not fully shift RNA with the highest protein concentration when compared to the full-length PURA and to PURA I-II.

(c) Do the human PURA I+II and dmPURA I+ II crystallize in the same space group and have similar packing? Can the observed structural flexibility be due to crystal contacts?

hsPURA I+II and dmPURA I+II crystallize in different space groups with different crystal packing. In both cases, the asymmetric unit contains 4 independent molecules with the flexible part of the structure composed of the β4 and β8 (β ridge) exposed to solvent. In the case of the *Drosophila* structure, we do not observe any flexibility of both β-strands. In contrast, for the human PURA structure the β ridge exhibits lots of flexibility and it adopts different conformations in all 4 molecules of the asymmetric unit. We observe similar flexibility of the β4 and β8 (β ridge) in the structure of K97E mutant which contains 2 molecules in the asymmetric unit. We would like to add that we expect crystal contacts to rather stabilize than destabilize domains.

Similarly, can the conformations observed for the K97E mutant be partially explained by packing?

Regarding the sequence shift observed for the β5 and β6 strands in hsPURA I+II K97E variant: although the β5 strand with shifted amino acid sequence is involved in the contact with the symmetry-related molecule with another β5 strand we don’t consider this interaction as a source of the shift. To be sure that the shift is not forced by the crystallization, we had performed NMR measurement which confirmed that in solution there is a strong change in the β-stands comparing WT and K97E mutant. This is an unambiguous indication that the structural changes observed in the crystal structure are also happening in solution. In addition, the MD simulations provide additional confirmation of our interpretation that K97E destabilizes the corresponding PUR domain. Taken together, we provide proof from three different angles that the observed differences indeed affect the integrity and hence function of the protein.

(d) Perhaps, it is my misunderstanding, but I find the NMR data on the Arg sidechains for the K97E confusing. If they are visible for K97E and not WT, doesn't this indicate that there is an exchange between two conformations or more dynamics in the WT structure? This does not seem to be the opposite of the expectation if K97E is thought to have more conformational flexibility.

Due to a technical issue (peak contour level), arginine side chain resonances were not clearly visible in the WT spectrum. The figure 5F has been updated. Now, they do correspond to those seen in the mutant spectrum. However, to prevent any confusion or mis/overinterpretation, we removed the sentence regarding arginine side chain: "Intriguingly, arginine side chain resonances Nε-Hε were only visible in the K97E variant, while they were broadened out in the wild-type spectrum."

(e) The most speculative part of the paper is the interpretation of SG and PB localization of PURA in Fig 1 and 2. There is an important issue with the statistics that must be clarified because it would appear that statistical significance was determined using each SG or PB as an independent measurement. This is incorrect and significance should be measured by only using the means of three biological replicates. This is well described here. It is not clear at this time if the reported P values will be confirmed upon reanalysis, and this may require reinterpretation of the data.

We are grateful for this clarifying comment and agree that the statistical analysis of P-body and stress granule was misleading. Of note, while the figures depicted all the values independent of the biological repeats, the statistical analyses were done on the mean value of each replicate of each cell line and not all raw data points.

We prepared new Plots, only showing the mean value of each replicate, and also re-calculated P-values. The values have changed only slightly in this new analysis because we now also included the previously labeled outliers (red points) to better demonstrate that significance still exists even when considering them.

In the new analysis of stress-granule association, only the value of the K97E mutant lost its significance, indicating that its association to stress granules is not lost. Therefore, we adjusted the following sentences in the manuscript.

Results:

Original: "While quantification showed a reduced association of hsPURA K97E mutant with G3BP1-positive granules (Fig 1B), the two other mutants, I206F and F233del, showed the same co-localization to stress granules as the wild type control."

Corrected: "In all the patient-related mutations, no significant reduction in stress granule association was seen when compared to the wild type control (Fig 1C)."

Original: "The observation that only one of the patient-related mutations of hsPURA, K97E, showed reduced stress granule association indicates that this feature may not constitute a major hallmark of the PURA syndrome. It should be noted however that this interpretation must be considered with some caution as the experiments were performed in a PURA wild-type background."

Corrected: "As we did not observe significant changes in the association of patient-related mutations of hsPURA to stress granules, it is suggested that that this feature may not constitute a major hallmark of the PURA syndrome. It should be noted however that this interpretation must be considered with some caution as the experiments were performed in a PURA wild-type background."

(f) A western blot showing the level of overexpression of the PURA proteins should be shown in Figure 1 as well as the KD of endogenous PURA for Figure S2?

As requested, a Western blot showing the level of overexpression of the different PURA proteins has been added as Appendix Figure S1.

A Western blot of the siRNA-mediated knock-down experiments of PURA and their corresponding control has been added to Appendix Figure S3. Quantification of three biological repeats showed a significant reduction of PURA protein levels upon knock down.

(g) While I appreciate that rewriting is time-consuming, I would recommend considering restructuring the manuscript because I think that it would aid the overall clarity. I think the foundation of the work is the structural characterization and would suggest beginning the paper with this data and the biochemical characterization. The co-localization with SGs and PBs and how this may be relevant to disease is much more speculative and is therefore better to present later. While I appreciate that the structural interpretation of why some mutants localize to PBs differently is not entirely clear, I do think that this would provide some context for the discussion.

In the initial version of the manuscript we first presented the structural characterization of PURA and afterwards the co-localization with SGs and PBs. As this reviewer stated him-/herself in (e), we also noticed that the SG and PB interpretation is the most speculative part of this manuscript. We felt that having this at the end of the results section would weaken the manuscript. On the other hand, we consider that the structural interpretation of mutations is much stronger and has a greater impact for future research. After long discussion we decided to swap the order to leave the most important results for the end of the manuscript.

**Recommendations for the authors: Reviewer #3**
Concerns and Suggested Changes:(a) For the characterization of G3BP1-positive stress granules in HeLa cells upon depletion of PURA, it remains unclear what is the efficiency of siRNA? The authors should provide a western blot to indicate how much the endogenous levels were reduced.

We completely agree with the stated concern and addressed it accordingly. We had performed this experiment prior to submission but for some unknown reason it was not included in the manuscript.

The Western blot of siRNA-mediated knock-down experiments of PURA and their corresponding control is now shown in Appendix Figure S3. Quantification of three biological repeats, showed a significant reduction of PURA protein levels upon knock down.

(b) How does knocking down PURA affect DCP1A-positive structures in HeLa cells? Would P bodies be formed even in the absence (or reduction) of total PURA?

Indeed, the stated question is very interesting. In fact, we have already shown in our recent publication (Molitor et al., 2023) that a knock down of PURA in HeLa and NHDF cells leads to a significant reduction of P-bodies. We actually referred to this finding on page 6:

"Since hsPURA was recently shown to be required for P-body formation in HeLa cells and fibroblasts (Molitor et al. 2023), PURA-dependent liquid phase separation could potentially also directly contribute to the formation of these granules."

On the same page, we also refer to the underlying molecular mechanism:

"However, when putting this observation in perspective with previous reports, it seems unlikely that P-body formation directly depends on phase separation by hsPURA, but rather on its recently reported function as gene regulator of the essential P-body core factors LSM14a and DDX6 (Molitor et al., 2023)."